



# Hydrology and runoff routing of glacierized drainage basins in the Kongsfjord area, northwest Svalbard

Ankit Pramanik[1,2,3], Jack Kohler[1], Katrin Lindbäck[1], Penelope How[4], Ward Van Pelt[5], Glen Liston[6], and Thomas V. Schuler[2]

[1]Norwegian Polar Institute, Tromsø, Norway
[2]University of Oslo, Oslo, Norway
[3]National Centre for Polar and Ocean Research, Goa, India
[4]Department of Remote Sensing, Asiaq Greenland Survey, Nuuk, Greenland
[5]Department of Geosciences, Uppsala University, Uppsala, Sweden
[6]Colorado State University, CO, USA

**Correspondence:** Ankit Pramanik (ankit.pramanik@outlook.com)

**Abstract.** Freshwater discharge from tidewater glaciers modulates fjord circulation and impacts fjord ecosystems. There can be significant delays between meltwater production at the glacier surface, and discharge into the fjord. Here, we present a hydrological analysis of the tidewater glaciers around Kongsfjorden, northwest of Svalbard, examining the pathways of glacier surface melt to the glacier fronts. To simulate discharge hydrographs at the outlets of the major drainage basins in the

Kongsfjord area we use 1) a simple, heuristic routing model and 2) the physically-based model HydroFlow to route runoff derived from a coupled surface energy balance – snow model. Plume observations at one of the tidewater glacier outlets and measurements of proglacial discharge of a land-terminating glacier are used for model calibration. Our analysis suggests that the local subglacial topography diverts a substantial amount of water from the drainage area of the glacier Kongsbreen to the neighboring glacier Kronebreen, across the border of their surface catchments. This is supported by the relative sizes of the

plumes observed at the respective glacier fronts. Runoff from the glaciers on the south side of the fjord is one order magnitude lower than runoff from the glaciers on the east and north sides of the fjord, reflecting differences in the size of the glaciers. We derive discharge hydrographs at all the major outlets of Kongsfjord basin, presenting here a detailed analysis of two of the glacierized basins. The average annual discharge period from the tidewater glaciers due to surface runoff was $105\pm10$ days. The largest discharge comes from Kronebreen, which is equivalent to around 40% of the total freshwater flux to the fjord.

## 1 Introduction

Freshwater discharge from tidewater glaciers has been shown to influence fjord circulation and the fjord ecosystem (Bartholomaus et al., 2013; Motyka et al., 2013; Lydersen et al., 2014; Fried et al., 2015; Urbanski et al., 2017; Everett et al., 2018). Meltwater generated at the glacier surface reaches the glacier front through crevasses, moulins, and englacial and subglacial channels, emerging at or near the base of the tidewater glacier front to create buoyant plumes. The upward movement of the

freshwater brings phytoplankton and zooplankton to the fjord surface, such that plumes can be active feeding grounds for birds and marine mammals (Lydersen et al., 2014). Fjord circulation is affected as the freshwater mixes at different vertical depths



of the fjord at the glacier front (Carroll et al., 2017). In contrast, runoff from land-terminating glaciers mixes with the fjord water at the surface, with only minimal effect on the local circulation, although the runoff still can influence the physical and chemical environment of the fjord (Nash and Moum, 2005; Nowak and Hodson, 2014, 2015). Recent studies have shown that

subglacial discharge controls the phytoplankton growth of the fjord (Halbach et al., 2019) and strongly correlates with the foraging behavior of the ringed seals in Kongsfjord, northwest Svalbard (Everett et al., 2018). The magnitude and timing of discharge peaks at the tidewater glacier front depends on the magnitude, timing and areal extent of the surface melt, and on the glacier's drainage system (Chandler et al., 2013; Dow et al., 2015), thereby influencing the occurrence of the foraging hotspots (Urbanski et al., 2017).

Water transport under glaciers generally occurs through a distributed drainage system, a channelized drainage system, or most typically, a combination of the two (Flowers, 2015). The distributed drainage system transports meltwater through a pervasive but inefficient system of pathways, in which pressures are close to the ice-overburden pressure. The channelized drainage is characterized by an arborescent system of relatively large and hydraulically efficient channels located over discrete parts of the glacier bed, in which pressures exceed ice-overburden less frequently (Paterson, 2013). During the winter and

early in the melt season, the relatively low discharge at the bed is accommodated through a distributed drainage system. With increasing meltwater amounts, a channelized drainage network develops, expanding at the expense of the distributed system.

Flow through the glacial drainage system is complex, with pathways that evolve rapidly in time, depending on the water flow and the glacier geometry (Röthlisberger, 1972; Walder, 1986; Kamb, 1987; Fountain and Walder, 1998; Flowers, 2015). Due to the inaccessibility of the glacier bed and a lack of observational data, it is difficult to accurately predict the nature of

time-varying subglacial hydrology and water routing of tidewater glaciers. A number of glacier hydrological models have been developed to calculate water flow through a coupled channel-distributed drainage system (Hewitt, 2011; Schoof et al., 2012; Hewitt, 2013; Schoof and Hewitt, 2013; Werder et al., 2013). These models have a large number of adjustable parameters, and generally require good quality input data and a high degree of computational power to model real-life glacier systems. A typical method to validate such models is by comparing modelled and observed discharge (Banwell et al., 2013). While water flow in

proglacial streams can be used to quantify discharge from land-based glaciers, measuring discharge directly at the termini of tidewater glaciers is not logistically feasible.

To first order, Water flow descends the hydraulic potential gradient which is determined by bed topography and basal water pressure, the latter is assumed to equal ice overburden pressure (Shreve, 1972). Here, we use steady-state hydraulic potential analyses to investigate drainage delineations and subglacial drainage networks under the tidewater glaciers of Kongsfjord basin

in Northwest Svalbard. To model discharge delays between the points where runoff is generated on the glacier and the major outlet points to the Kongsfjord basin, we use the modeled runoff (Pramanik et al., 2018) from a coupled energy balance-snow model (Van Pelt et al., 2012; Van Pelt and Kohler, 2015) in a simple routing model, as well as a more physically-based model, HydroFlow (Liston and Mernild, 2012). Data on plume area extent of a tidewater glacier, and a time-series of proglacial discharge on a land-terminating glacier are used to calibrate the routing model. Plume surface area has been used previously

as a proxy for discharge (Mankoff et al., 2016; Schild et al., 2016; Slater et al., 2017), and recent studies in Greenland have





inferred information about subglacial hydrology through a combination of plume observations and modelling (Banwell et al., 2013; Carroll et al., 2015; Slater et al., 2017).

## 2  Study area

Kongsfjord is the site of a number of interdisciplinary research activities addressing the importance of the freshwater influx to
the fjord (Sundfjord et al., 2017; Everett et al., 2018; Halbach et al., 2019; Pramanik et al., 2019). The fjord is surrounded by both tidewater and land-terminating glaciers of varying size, whose discharge impacts the fjord circulation and biogeochemistry (Cottier et al., 2005; Halbach et al., 2019). Warm water inflow from the Gulf Stream has in recent years kept the western fjords of Svalbard almost ice-free year-round, and Kongsfjord has until recently not had significant sea ice cover, since 2006 (Negrel et al., 2018).

65       Kongsfjorden is open to the ocean to the west and surrounded by glaciers on the other three sides: Glaciers on the south side are land-terminating, whereas those to the east and north are mostly tidewater glaciers. Two fast-flowing tidewater glaciers, Kronebreen and Kongsbreen, are fed by the icefields Holtedahlfonna and Isachsenfonna, respectively. Kongsbreen has two termini, Kongsbreen North and South, which are divided by the bedrock ridge Ossian Sarsfjellet. The other tidewater glaciers are Kongsvegen, Conwaybreen, and Blomstrandbreen (Fig.  1). The terminus depths of these tidewater glaciers are relatively
shallow, with maximum depths of ca. 120 m.

     Previous glacio-hydrological studies in the Kongsfjord basin have investigated the subglacial hydrology of the lower part of Kronebreen (How et al., 2017), while Lindbäck et al. (2018) mapped the subglacial topography of the upper catchment, Holtedahlfonna and Isachsenfonna, in the context of future studies of glacier dynamics, hydrology, geology and fjord circulation. These two adjacent icefields are the largest in this region, and contribute the largest amounts of freshwater to the fjord
(Pramanik et al., 2018).

     The total drainage area of the Kongsfjord basin is roughly 1440 $km^2$, 80% is glacier-covered. Previous modelling indicates that glaciers on the eastern side of the fjord contribute most of the freshwater to the fjord, and runoff from seasonal snow in the non-glacierized area contributes only 16% to the total runoff (Pramanik et al., 2018). Kongsfjord basin has a few rivers and streams on the south side originating from the land-terminating glaciers, which drain through proglacial valleys into
the fjord. One of the major rivers is Bayelva, which receives runoff contributions from the two small glaciers Austre and Vestre Brøggerbreen and the surrounding glacier-free terrain. Four glaciers in the Kongsfjord basin are monitored by the Norwegian Polar Institute: Austre Brøggerbreen, Midtre Lovénbreen, Kongsvegen and Kronebreen-Holtedahlfonna (Fig.  1) (Kohler, 2013).



## 3  Data

### 3.1  Runoff from a Coupled energy balance-snow model

A coupled surface energy balance - snow model (Van Pelt et al., 2012) has been used previously to quantify runoff from the entire Kongsfjord basin (Pramanik et al., 2018). The surface energy balance model simulates melt by considering all the energy fluxes at the glacier surface. The energy balance model is connected to a vertical subsurface model, which simulates subsurface temperature, density, and water content. Meltwater percolates through the snowpack, and may refreeze depending upon the subsurface temperature and density, or maybe stored as irreducible water (depending upon the available pore space). Residual meltwater that reaches the base of the porous snow- or firnpack is assumed to be available as runoff. Runoff from the entire Kongsfjord basin was simulated over the period 1980-2016 in 6-hour intervals on a 250-m grid (Pramanik et al., 2018). The average simulated runoff for the individual summer months during 2013-2016 is shown in Fig. A1. Here, we are interested in discharge over the period 2010-2016, when extensive biological and oceanographic data were collected (Hamilton et al., 2016; Everett et al., 2018).

### 3.2  Surface and bed digital elevation models

We use a 5-m gridded digital elevation model (DEM) of the glacier surface, constructed from aerial photographs taken in 2009 and 2010 (Norwegian Polar Institute, 2014), which is resampled onto a 250-m grid (Fig. A2a). The basal topography of the tidewater glaciers in the Kongsfjord basin has been mapped previously with airborne and ground-based ice-penetrating radar (Lindbäck et al., 2018) to derive a 150-m bed DEM, which is resampled to the same 250-m grid as the surface DEM (Fig. A2b).

### 3.3  Plume data

Plume observations were made with a time-lapse camera over the period 2014-2016. The camera was installed on Collethøgda, a mountain between the termini of Kronebreen and Kongsbreen, and took pictures at every hour. The extent of the plume at the fjord surface was digitized manually from cloud-free images, as described in How et al. (2017). Subsequently, plume extents were georectified with the PyTrx toolset as outlined in How et al. (2020), using a set of ground control points, camera model, and the surface DEM described previously (Norwegian Polar Institute, 2014).

### 3.4  Bayelva discharge data

The total area of the Bayelva basin is 33 $km^2$, of which 17 $km^2$ is glacierized. In early summer, discharge is from snowmelt runoff of the non-glacierized area, and runoff of snow and ice melt from glaciers. In late summer, rainfall and runoff from the glaciers contribute to water flow in the river. Discharge measurements at the Bayelva station (Fig. 1) are conducted automatically; the water level in a concrete-floored weir is measured at hourly intervals by a pressure transducer, and a float and wire system (Killingtveit et al., 2003). The system is calibrated periodically to derive a rating curve that converts water





level to discharge. Over the period 1990-2000, only daily data are available, whereas over the period 2000-2015 data were

stored hourly. Because the monitoring is unattended, the discharge data have periods with erroneous readings, caused by ice or sediment build-up at the sensor; however, the timing of discharge events (i.e. the peaks) is in general not affected.

## 4 Method

### 4.1 Flow paths and drainage basins

We use flow algorithms in the MATLAB package Topotoolbox (Schwanghart and Scherler, 2014) to determine flow direction

and drainage basins. The land-terminating glaciers are constrained within relatively narrow valleys, and observations show that runoff routing from these glaciers is controlled predominantly by surface topography alone for most of the glacier area, therefore, we use the surface DEM to evaluate flow directions and drainage basins. The tidewater glaciers are thicker (Lindbäck et al., 2018) and runoff routing is influenced by both surface and subglacial topography (Vallot et al., 2017).

We compute hydraulic potential for tidewater glaciers to determine water routing (Lindbäck et al., 2015; Everett et al., 2016).

Hydraulic potential is the sum of the basal water pressure and elevation potential; the former is approximated as a fraction, $k$, of the ice overburden pressure via:

$$\phi = P_w + P_e = k\rho_i gH + \rho_w g(Z_i - H) \tag{1}$$

where $\rho_i$ is the density of ice (916 kg-$m^{-3}$), $\rho_w$ is the density of water at 0°C (1000 kg-$m^{-3}$), $g$ is the gravitational acceleration in m-$s^{-2}$, $H$ is the ice thickness in m, and $Z_i$ is the ice surface elevation in m (Shreve, 1972). The hydraulic head,

$h$ (in metres), can be expressed as follows:

$$h = k(\frac{\rho_i}{\rho_w})H + (Z_i - H) \tag{2}$$

When $k$ equals 1, the subglacial water pressure is equivalent to the ice overburden pressure, a situation we may expect in winter, in the absence of low-pressure channels (Zwally et al., 2002). If $k$ equals 0, the subglacial water is at atmospheric pressure and water moves according to the bed topography alone. To calculate hydraulic potential, we assume a spatially and

temporally constant value of $k$, and route water accordingly.

We use the 250-m grids of surface and bed elevations to derive the hydraulic potential, and calculate subglacial drainage networks with Topotoolbox (Schwanghart and Scherler, 2014). The model calculates the slope of each grid cell, with respect to the adjacent eight grid cells, and water is assumed to flow along the steepest direction. Depressions in the hydraulic potential surface are filled to remove sinks. Grid cells that are not receiving any flow from upstream lie on the boundary of the basin.

Each basin has a single outlet where all the grid cells of the basin drain, such that these grid cells demarcate the respective drainage basin area. We investigate the effect of changes in water flow, and thereby the catchment areas, by varying $k$ values between 0 and 1.

The hydraulic potential may depend on the resolution of the DEMs, where uncertainties in hydraulic potential increase with the coarser resolution. Therefore, we further calculated hydraulic potential for different DEM resolutions of 100-m and 150-m.





To check the robustness of the analyses, we conducted 10000 Monte-Carlo simulations in each cases of hydraulic potential calculation; i) with spatially distributed random $k$ values and ii) with randomly perturbed ice thickness.

## 4.2 Runoff routing models

We use two approaches: the first uses a simple routing model where water moves at uniform speed to calculate the discharge hydrograph at the outlet points of each catchment. The second routing approach uses a topographically-controlled linear reservoir
model, HydroFlow, to calculate discharge hydrographs at the outlet points of the studied glaciers.

### 4.2.1 Simple routing model

The main idea behind the simple routing model is that distance and water wave speed are the first-order factors affecting runoff routing (Kohler, 1995). While the wave speed is known to be a function of many factors, including surface slope, water storage, presence of crevasses and moulins, and seasonal changes in supraglacial, englacial, subglacial channel dimensions and
roughness, we here apply a uniform and constant water movement speed to calculate the discharge hydrograph at the outlet points of all the drainage basins (Cowton et al., 2013; Slater et al., 2017).

Time-varying runoff from the energy balance - snow model (Sect. 3.2) is used as the input to the routing model using 250 m gridded DEM of the surface and bed elevation. We use the hydraulic potential (calculated in Sect, 4.1) to derive flow paths and the distance of each grid cell to its basin outlet point. We assign a uniform water speed to calculate a time delay associated
with each grid cell, and sum the delayed runoff from all the grid points in an individual glacier basin to derive the discharge hydrograph at its outlet point. We test different wave speeds $(0 - 1 \text{ m-}s^{-1})$ with intervals of 0.1 m-$s^{-1}$ to consider the effect of the slower or faster flow of water, and use observational plume and discharge data to determine an optimal wave speed.

The primary factor controlling plume generation is runoff from glaciers. Plume area extent depends on several factors (such as wind speed and direction, terminus depth, discharge, and fjord stratification and circulation), and there does not appear to be a
simple correlation between the discharge and plume area on a seasonal timescale (How et al., 2017). Our main assumption here is that the timing of maximum plume extent coincides with maximum discharge at the front. We derive discharge hydrographs at Kronebreen outlet for different water speeds, and compare the peaks in the delayed runoff to the peaks in plume extent. In the absence of any direct discharge measurements at the tidewater glacier front, the water speed for the Kronebreen glacier runoff route is calibrated by computing the normalized cross-correlation between the high-frequency components of the modelled
discharge and plume area data (Fig. B1, discussed in detail in the Appendix B).

We also compare the discharge hydrograph at Bayelva to that modelled in the Brøggerbreen glacier catchment. A single general value for the water speed of the simple routing model is tuned for each of the cases; subglacial water routing of tidewater glaciers and supraglacial water routing for land-terminating glaciers. For Bayelva, the Nash-Sutcliffe coefficient (NSC) is calculated over the year 2000-2010, quantifying the agreement between the modelled and observed hydrographs. We
calculate the time difference between the peaks of no-delay and delayed discharge, thereby referred to as delay-time, which is governed by the assumed value of the water speed and the distance.





### 4.2.2 HydroFlow model

The second routing model, HydroFlow, uses a linear-reservoir approach to calculate discharge hydrographs at the outlet points of the glaciers (Liston and Mernild, 2012). HydroFlow has been applied previously in different glacierized and river catchments
(Mernild et al., 2015, 2017). It was developed by Liston and Mernild (2012) and first tested in the Mittivakat glacier catchment in southeast Greenland. The model is described in detail by Liston and Mernild (2012), and a brief summary is given below.

HydroFlow is a gridded linear reservoir model, where each grid cell acts as a linear reservoir, transferring water to the steepest adjacent cell. The routing is conducted in four routines. First, the model calculates the topographically controlled flow networks from the surface DEM. Second, it calculates the individual watersheds inside the domain of interest. In the third part,
it assumes that there are two different components associated with water transport: a slow-response and a fast-response system. The slow time-scale accounts for distributed runoff through the grid cell matrix, for example flow within the snow, ice, and soil, whereas the fast time-scale represents water transport through the channelized system and includes supra-glacial, sub-glacial, or en-glacial flow. Here, we used hydraulic potential instead of surface DEM to make water flow according to the hydraulic potential gradient for tidewater glaciers. All parameters of the model in this study are adopted from Liston and Mernild (2012),
except the timescale $\alpha$ associated with the fast-flow, which we calibrate, as described in detail below.

HydroFlow is a linear-reservoir model and does not consider explicitly subglacial hydrology in its flow routine. For land-terminating glaciers of this region, supraglacial channels play a major role to transport water, but for tidewater glaciers, sub-glacial hydrology is important. We calibrated the fast-time scale coefficient $\alpha$ of the HydroFlow model to derive discharge hydrographs for Bayelva and Kronebreen, following a similar procedure for the simple routing model (sect. 4.2.1); we com-
pute NSC and the normalized cross-correlation between modelled and measured discharge for Bayelva, and for high-frequency components of the modelled discharge and observed plume for Kronebreen, respectively. Two separate coefficients were de-rived for the tidewater and land-terminating glaciers.

## 5 Results

### 5.1 Subglacial hydrology and drainage basins

Based on the surface DEM, there are 114 basins draining into Kongsfjorden, of which 15 are either completely or partially glacierized, and are named after the respective glaciers (Fig. 1). From the hydraulic potential at the base of the tidewater glaciers, there are five major subglacial catchments with an area of $50 \, \mathrm{km}^2$ or larger (Table A1). We vary $k$ values between 0 and 1 to find that the subglacial drainage delineation deviates from the outlines of the surface catchments at the ablation zone of the glacier, implying subglacial transfer of water across the boundaries of surface catchments, a phenomenon referred to as water
piracy (Anandakrishnan and Alley, 1997; Lindbäck et al., 2015). Water piracy is apparent between the two largest adjacent ice fields of the basin, Holtedahlfonna and Isachsenfonna, which would drain to Kronebreen and Kongsbreen, respectively, according to the surface flow. We find that for a $k$ value between 0.5 and 1, a substantial part of Isachsenfonna drains to Holtedahlfonna subglacially, whereas the reverse happens for $k$ values between 0 and 0.1 (Fig. A3). We also found that for





$k$ values between 0 and 0.2, Kronebreen drains through an outlet at the southern side of the glacier which merges with the
Kongsvegen outlet and drains as a single outlet point (Fig. A4, A5). Monte-Carlo simulations of hydraulic potential with
spatially distributed random $k$ values show around 94% of water piracy cases are from Isachsenfonna to Holtedahlfonna,
whereas only 5% of cases are from Holtedahlfonna to Isachsenfonna, and below 1% of cases occur with no water piracy
between these two glaciers (discussed in detail in Appendix Sect. A2). Fig. 2 shows the changes in drainage catchment for
surface DEM and for hydraulic potential corresponding to $k = 0.1, 0.5,$ and $0.9$, representing substantial changes in drainage
basins. For $k$ values between 0.5 and 1, a substantial amount of water from Isachsenfonna flows to Kronebreen. Although the
catchment area of Kongsbreen decreases by 75% for this water piracy, the total area is still $82 \text{ km}^2$, the fourth largest catchment
in this area. This water piracy affects the water influx to the fjord from Holtedahlfonna and Isachsenfonna in late summer only,
since the switching between Holtedahlfonna and Isachsenfonna takes place in a small area situated in the upper part of these
two glaciers, where meltwater is only produced in late summer. Because of this, water influx to the fjord through the outlets of
these two catchments will differ substantially only in peak summer (July-August), when runoff occurs from the upper part of
the glaciers.

The drainage areas of the two other tidewater glaciers, Conwaybreen and Blomstrandbreen, are not affected by the choice of
$k$ as these two glaciers are constrained in well-defined valleys. The drainage area of Kongsvegen changes only for $k$ values of
0.1 to 0.2, as the lower part of Kongsvegen abuts Kronebreen to only a small degree.

**5.2 Discharge hydrograph**

Normalized cross-correlation is calculated for different water speeds of the simple routing model and for different $\alpha$ values
of the HydroFlow model, comparing the discharge and plume area data (Fig. B2). From the visual inspection of Normalized
cross-correlation, we found the best fits to the data for water speeds of $0.6 \text{ ms}^{-1}$, and for $\alpha$ values of 0.2, which are used for the
final model run. The same water speed of $0.6 \text{ ms}^{-1}$ is also optimal for Bayelva, yielding the highest Nash-Sutcliffe coefficient
in all years (Fig. B3a). However, the best $\alpha$ values, as determined by the highest Nash-Sutcliffe coefficients are 0.15 and 0.2
(Fig. B3b). We observe, however, that $\alpha = 0.15$ results in a smoothed hydrograph, whereas $\alpha = 0.2$ gives sharper peaks which
better resemble the measured discharge peak. We therefore choose $\alpha = 0.2$ as the optimal value.

Fig. 3a shows the major glacierized drainage basins ($>10 \text{ km}^2$) and their respective outlet points, where the subglacial
drainage delineation of the tidewater glaciers was derived assuming $k = 0.8$. The hydraulic potential analysis shows a prevalence
of water piracy from Isachsenfonna to Holtedahlfonna occurred for $k$ values between 0.5 and 1, therefore, we use the hydraulic
head for $k = 0.8$ (the average of the $k$ value range) to derive water routing. Fig. 3b shows the distributed delay map associated
with each grid cell in the domain, assuming a water speed of $0.6 \text{ ms}^{-1}$. The runoff time-delay for the land-terminating glaciers
of the southern side of the fjord is within the time resolution of the runoff model (6 hours), while the delay from upper parts of
the tidewater glaciers reaches up to a day.

Fig. 4 shows the discharge hydrograph for the Kronebreen outlet derived from the simple routing model and HydroFlow over
the period 2013-2016. We find that the discharge peaks of both the models coincide with each other, whereas the HydroFlow
hydrograph is smoother than the simple routing model. The average number of discharge days over a year from all the five





tidewater glacier outlets over 2013-2016 period was 105±10 days, with some late discharge events in 2016 mostly due to rain events. We found average time delays of around 17-30 hours for high discharge events (>150 $m^3 s^{-1}$) due to drainage of the
upper part of the tidewater glaciers.

Fig. 5 shows the hydrographs derived by simple routing model and HydroFlow compared to the measured Bayelva discharge. We calculated NSC and correlation for Bayelva between the observations and results of the simple routing model and HydroFlow. The comparison is carried out on the summer months only, between 15 May and 15 October, however, for 2014 the comparison spans 4 July to 15 October as the Bayelva observations start from 4 July only. Comparison of the modelled and
measured data shows good agreement, with average NSC values of 0.54 and 0.65 for the simple routing model and HydroFlow, respectively. The discharge data of 2015 are not complete and 2016 is not available. We also compare peak discharge between the modelled and measured hydrographs to evaluate model performance for Bayelva. The simple routing model hydrograph peaks match well with the measured discharge with significant correlation of 0.92, indicating that the simple routing model is able to realistically reproduce delays and hence discharge on sub-daily scales.

Fig. 6a shows the annual runoff volumes from the land-terminating glaciers larger than 10 $km^2$ on the south side of the fjord for the period 2013-2016. The Bayelva basin contributes the largest amount of freshwater from the south side of Kongsfjord. Fig. 6b shows the annual runoff volumes of the six major (drainage area >50 $km^2$) outlet points comprised of tidewater glaciers on the east and north sides of the fjord for the year 2013-2015. We simulate the discharge hydrographs for all the drainage sub-basins of the Kongsfjord basin; however, we discuss only the results of the discharge hydrographs for one tidewater
glacier (Kronebreen) and one land-terminating glacier basin (Bayelva). The maximum amount of freshwater comes through Kronebreen, comprising 39±1% of the total discharge to the fjord. Annual runoff from tidewater glaciers is about one order of magnitude greater than that from land-terminating glaciers, and is corresponding to their area proportion.

## 6 Discussion

### 6.1 Hydrology

The Kronebreen-Holtedahlfonna and Kongsbreen-Isachsenfonna surface drainage areas are 421 and 337 $km^2$ respectively, but a substantial change in the subglacial catchment area occurs for $k$ values around 0.5 and above, resulting in the upper part of Kongsbreen-Isachsenfonna draining to Kronebreen. This means modelled Kronebreen discharge is around four times higher (4.46±0.33 for 2013 -2016 runoff) than that of Kongsbreen, whereas discharge at these two glacier outlets is comparable when flow is governed by surface topography alone. An opposite process occurs between these two glaciers for $k$ values between
0 and 0.4, in which the upper part of Kronebreen-Holtedahlfonna drains to Kongsbreen. This would result in Kongsbreen discharge being around five times higher than Kronebreen discharge. The potential subglacial water capture from Isachsenfonna to Holtedahlfonna is supported qualitatively by satellite and terrestrial time-lapse imagery (Fig. A6), which consistently shows a significant plume in front of Kronebreen (How et al., 2017), but only a small plume at Kongsbreen (Schild et al., 2018). In addition, mammals and birds are observed to forage preferentially at the front of Kronebreen (Lydersen et al., 2014; Urbanski
et al., 2017), an indication of vigorous upwelling induced by subglacial discharge.




Our model depends on the simplifying assumption of constant, uniform $k$, whereas subglacial water pressure varies in time and space. However, the Monte-Carlo simulations using spatially distributed random $k$ values, and the observational data, both support the hypothesis that the water is diverted from the Isachsenfonna catchment to the Holtedahlfonna-Kronebreen catchment. How et al. (2017) investigated the subglacial hydrology of lower part of Kronebreen, and found the locations of the

Kronebreen outlet shifts depending on the $k$ value used, with its position to the north of the front for $k > 0.6$ and to the south for $k < 0.6$. Our findings are similar but with a slightly different threshold $k$ value of 0.5. A possible reason for discrepancy is likely the different DEM resolutions (Fig. A3- A5). Hydraulic potential and corresponding drainage delineations are resolution dependent, where a higher resolution analysis is expected to better represent the actual scenario. Repeating the same analysis with different resolution of DEMs, we found that the threshold $k$ values that causes the change in subglacial hydrology structure

shifts a little for different DEM resolution (Fig. A3- A5). However, all the analyses at different resolutions still reproduce the same piracy between Isachsenfonna and Holtedahlfonna.

To further examine the question of subglacial water piracy, simulations using a process-resolving model of subglacial hydrology are required which more realistically elucidates spatio-temporal evolution of subglacial water pressure and associated hydraulic potential. Frequent and continuous monitoring of plume area extent at those two outlet glaciers is essential to infer

more information about time-evolving subglacial drainage as long as no other reliable technique to monitor plume discharge is established. The remaining glaciers in the Kongsfjord basin are well constrained by topography and their subglacial drainage areas are insensitive to changes in $k$ values. Further work could investigate in more detail the evolution of the subglacial drainage networks, especially under the Isachsenfonna and Holtedahlfonna icefields. Subglacial water pressure measurement through a network of boreholes could give more insight into the subglacial hydrology of this sensitive zone, such as studies

carried out on Store Glacier in Greenland (Doyle et al., 2018) and Glacier Perito Moreno in Patagonia (Sugiyama et al., 2011).

## 6.2   Discharge hydrograph

The subglacial hydrology of glaciers and ice sheets is difficult to model due to the complexity of the processes, and the relative lack of data at the bed. A number of hydrological models with different degrees of complexity have been developed (Hewitt, 2011; Werder et al., 2013; Bueler and Van Pelt, 2015; Flowers, 2015). It is an ongoing debate among hydrological modelers

whether a higher degree of complexity improves model accuracy, especially in predicting freshwater discharge at tidewater glacier outlets (Li et al., 2015; De Fleurian et al., 2018). Here we are mainly interested in calculating discharge hydrographs at the tidewater glacier outlets on sub-daily timescales. We argue that the simple routing model allows adequate quantification of the discharge hydrographs at outlet points around the Kongsfjord basin, given that the main intent is to derive time delays at the outlets. The discharge hydrographs derived from the simple routing scheme compares well to those from the more physically

realistic linear-reservoir model HydroFlow. We used plume data as a proxy to discharge to calibrate the parameter of the simple routing model. We found a little difference in the time period of discharge and plume data for Kronebreen for one year. In 2014, some plume activity was visible in late September, even though the simulated runoff was close to zero. This late plume activity could be caused by runoff from basal melting or from a late rain event occurring at lower elevations, not captured by the model (Pramanik et al., 2018, 2019). Kronebreen is polythermal and fast-flowing glaciers, thus there can be substantial basal melting





in early summer as well as in late autumn. The shallow water depth of Kronebreen front makes the runoff, even with low magnitude, appear at the surface.

Although we assign an optimum water speed for the routing model, we propose that the water wave speed will play a crucial role when discharge is calculated over a sub-hourly timescale. We did not try to find an accurate water speed value as our model presents the complex physical process through simple parameterization. Our discharge hydrographs are relatively less sensitive

to the choice of water speed due to the relatively coarse temporal resolution of the modelled discharge data. We observe that a simple routing model performs equally well with a physically-based linear-reservoir model when discharge is simulated over sub-daily timescale. However, we suspect that a major difference between a single parameter simple routing model and a complex model will be apparent when simulation is done over sub-hourly timescale, as complex models are expected to capture details of the hydrology and water routing. However, a complex model may even fail to produce an accurate discharge

hydrograph due to a lack of temporal observational data from the subglacial environment. In all these models, uncertainty comes in simulating discharge timing, but the magnitude of discharge remains unaffected. Nonetheless, with a lack of observational data, a simple model serves the basic purposes of biogeochemistry and oceanographic studies (Sundfjord et al., 2017; Everett et al., 2018; Schild et al., 2018; Halbach et al., 2019).

The five tidewater glaciers of the Kongsfjord basin contribute most of the freshwater flux to the fjord. This has significance

as mixing of freshwater from glaciers with the fjord seawater influences circulation (Sundfjord et al., 2017), and enhances primary and secondary production, thus playing an important role in the fjord ecosystem (Lydersen et al., 2014).

Here, we used a time-independent modelling approach to route meltwater to the outlet. We assume stationary and uniform values for hydraulic potential, water travel speeds and fast-response timescale for the entire season. However, in nature, these values would depend on several factors e.g., evolution of channels, surface slope, bedrock property, and vary temporally.

Further uncertainty comes from calibration using plume area observations as a proxy for subglacial discharge. Plume extent is not a direct signal of outflow, and the routing model could further be improved by incorporating more time-dependent physical processes. Future studies could combine detailed hydrology and routing in a model framework to get a more robust estimate of discharge at the glacier outlet points.

## 7 Conclusions

Freshwater influx from glaciers substantially impacts fjord circulation and fjord ecosystem. Glacier hydrology and water routing play the major role in controlling discharge hydrographs at fjord inlets. We analysed the subglacial hydrology and water routing of the entire glacierized area of the Kongsfjord basin in order to simulate discharge hydrograph of freshwater influx at different inlets points of the fjord. Subglacial hydrology of this region is poorly understood, and here, using steady-state hydraulic potential analyses, we hypothesize a structure of the subglacial environment of the basin's tidewater glaciers. We

suggest that there is a higher possibility of subglacial water piracy between Isachsenfonna and Holtedahlfonna, where the latter receives substantial water subglacially from the former. Our hypothesis is supported by the relative size of the plumes at the two tidewater glacier fronts. Furthermore, we implement a simple routing model to derive discharge at the different drainage





catchments around Kongsfjord. The discharge hydrograph of simple routing model is compared with the ones derived from Hy-
droFlow. We conclude that, with lack of observation data of subglacial conditions, the simple routing model performs equally

well with HydroFlow in simulating discharge hydrographs over sub-daily timescales.

## Appendix A: Hydrology

### A1 Uncertainty

To determine the uncertainty in the hydraulic head, we estimated the uncertainty $\sigma_h$ using standard analytical error propagation
method. The uncertainty in $\sigma_h$ is calculated as

$$\sigma_h = \sqrt{(\frac{\delta h}{\delta H})^2(\sigma_H)^2 + (\frac{\delta h}{\delta Z_i})^2(\sigma_{Z_i})^2 + (\frac{\delta h}{\delta \rho_i})^2(\sigma_{\rho_i})^2 + 2(\frac{\delta h}{\delta H})(\frac{\delta h}{\delta Z_i})(\sigma_{HZ_i})^2} \quad (A1)$$

Where $\sigma_h$ is standard deviation in ice thickness $H$, $\sigma_{Z_i}$ is the standard deviation in surface DEM $Z_i$, $\sigma_{\rho_i}$ is the standard
deviation in the density of ice $\rho_i$, $\sigma_{HZ_i}$ is the covariance of $H$ and $Z_i$. The uncertainty in ice thickness DEM was calculated
to be ±24 m (Lindbäck et al., 2018). The surface elevation dataset has a standard uncertainty of 2-5 m (Norwegian Polar
Institute, 2014). The uncertainty of the density of water is very small, hence neglected. The combined estimated uncertainty in

the hydraulic head is ±22 m.

### A2 Sensitivity (Monte Carlo)

To check the robustness of hydraulic potential analyses, we conducted sensitivity tests with different DEM resolution and
monte-carlo simulations with randomly distibuted k values and with random Ice thickness perturbations.

#### A2.1 Different DEM resolution

We conducted the hydraulic potential analysis for bed and surface DEM of 150-m and 100-m resolution to check the depen-
dency of drainage delineation and water piracy results on the DEM resolution, and applicability of the analyses in routing
model. We found that the drainage delineations are little sensitive to resolutions, however, the water piracy between Holtedahl-
fonna and Isachsenfonna is aptly represented in all the DEMs (Fig. A3- A5).

#### A2.2 Random spatially distributed k values

We did 10000 Monte-Carlo simulations for hydraulic potential with spatially distributed random $k$ values ranging between 0
to 1. Thereby, we investigated the changes in subglacial drainage delineations. We found 94.36% cases where Isachsenfonaa
is draining to Kronebreen, 0.38% cases with no subglacial water piracy and 5.25% cases where Holtedahlfonna is draining to
Kongsbreen-North. Between Kronebreen and Kongsvegen, we found 29.72% cases where Kongsvegen and Kronebreen drain
separately whereas rest 70.28% cases show a single outlet for Kronebreen and Kongsvegen.

### A2.3   Random noise in Ice thickness (with standard deviation)

We also did 10000 monte-carlo simulations for hydraulic potential with random DEM perturbations. We perturbed the Ice thickness by adding random noise within the error of Ice thickness measurement, which is $\pm24$ m (Lindbäck et al., 2018). We found that in all the runs, Isachsenfonna drains to Holtedahlfonna, thus shows that the water piracy between Holtedahlfonna and Isachsenfonna is not sensitive to the DEM perturbation within the error limit of bed DEM.

### Appendix B:  Routing

### B1   Calibration of parameters of Routing model

We use a median filter of 60-hours to filter out the low-frequency part of the signals from both modelled discharge and plume data (Fig. B1). Thereafter, we calculate normalized cross-correlations between the high-frequency part of the plume and modelled discharge for different water speeds. We found the least lag between modelled discharge and plume data for the water speed value of $0.6$ ms$^{-1}$, and use this value for the final model run. We choose a cut-off frequency to filter out low-frequency components and compare the high frequency of plume with the high frequency of discharge data. We did not find good correlation of plume and discharge data for all cut-off frequency. We used different cut off frequencies and calculated normalized cross-correlation. A good correlation is observed for the cut-off frequency of 59 hours and above. Therefore, we use that as the final cut-off frequency to extract high-frequency signal to calibrate the water speed of the routing model.

*Author contributions.* AP and JK conceived the idea. AP conducted all the analysis and wrote the manuscript. JK and AP wrote the simple routing model script. KL provided the bed elevation data, PH provided plume data of Kronebreen, and GL provided the HydroFlow model code. All authors took part in discussion and editing of the paper.

*Competing interests.* The author declares that no competing interests are present.

*Acknowledgements.* The first author received Ph.D. studentship support from National Centre for Polar and Ocean Research (NCPOR), financed by the Ministry of Earth Sciences, Government of India (Grant/Award number: MoES/16/22/ 12-RDEAS (PhD fellowship-NPI)). Additional funding came from Polish-Norwegian Research Programme GLAERE, the Norwegian Research Council project TIGRIF, and the Norwegian Polar Institute's TW-ICE project.



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

**Figure 1.** The Kongsfjorden study area in Svalbard showing tidewater glaciers (orange), land-terminating glaciers (red) and non-glacierized areas (green). Bayelva River (blue polyline), Bayelva discharge measurement site (white dot), and Ny-Ålesund (pink triangle) are also shown in the southwest side of the basin. The figure includes Landsat mosaic image and contours from a digital elevation model (Norwegian Polar Institute, 2014).







**Figure 2.** Subglacial Drainage basins of tidewater glaciers for a) surface drainage, b) $k = 0.1$, c) $k = 0.4$, and d) $k = 0.9$. Drainage basins are named with respective glaciers. Kronebreen basin is marked with blue, Kongsbreen basin with orange, Kongsvegen basin with cyan, Conwaybreen basin with maroon, and Blomstrandbreen basin with green. Basin outlets are shown with magenta circle. Note that the streams covering more than 50 grid cells are shown here.



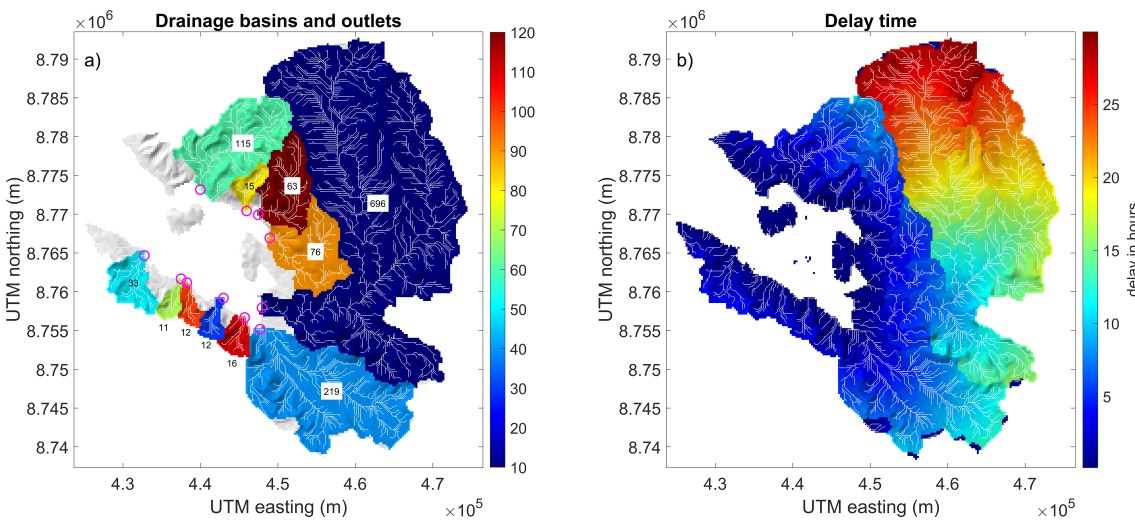

**Figure 3.** a) Drainage basins associated with glaciers (drainage area >10 km$^2$) and their outlet points. The associated drainage area (km$^2$) is shown for each basin. b) Delay times associated with each grid cell, based on the simple delay model.





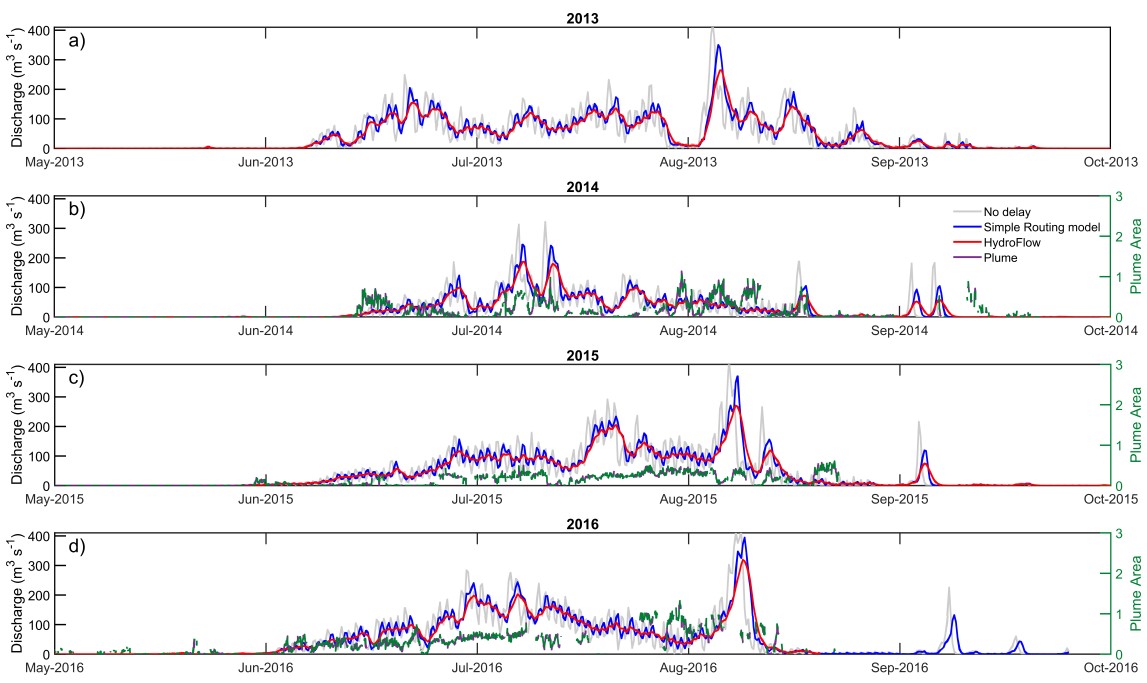

**Figure 4.** Discharge hydrograph at Kronebreen outlet for a) 2013, b) 2014, c) 2015, and d) 2016. The plume area extent is shown in dark green color. Note that plume observation starts from 2014 only.



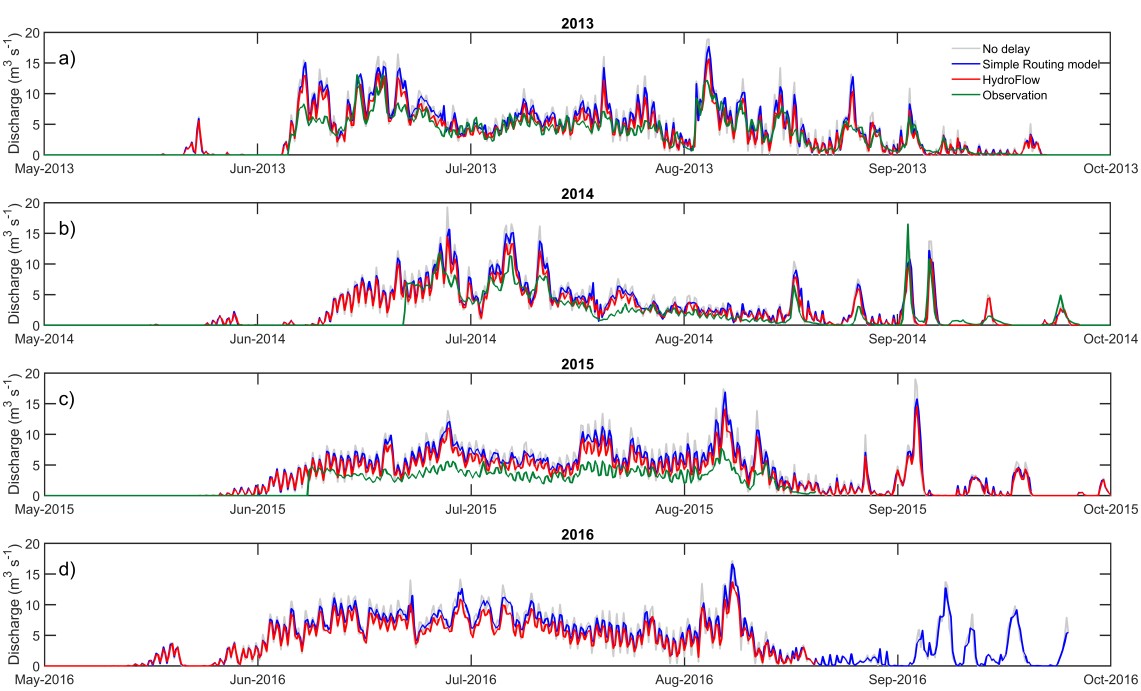

**Figure 5.** Bayelva measured and modelled discharge hydrograph for a) 2013, b) 2014, c) 2015, and d) 2016.





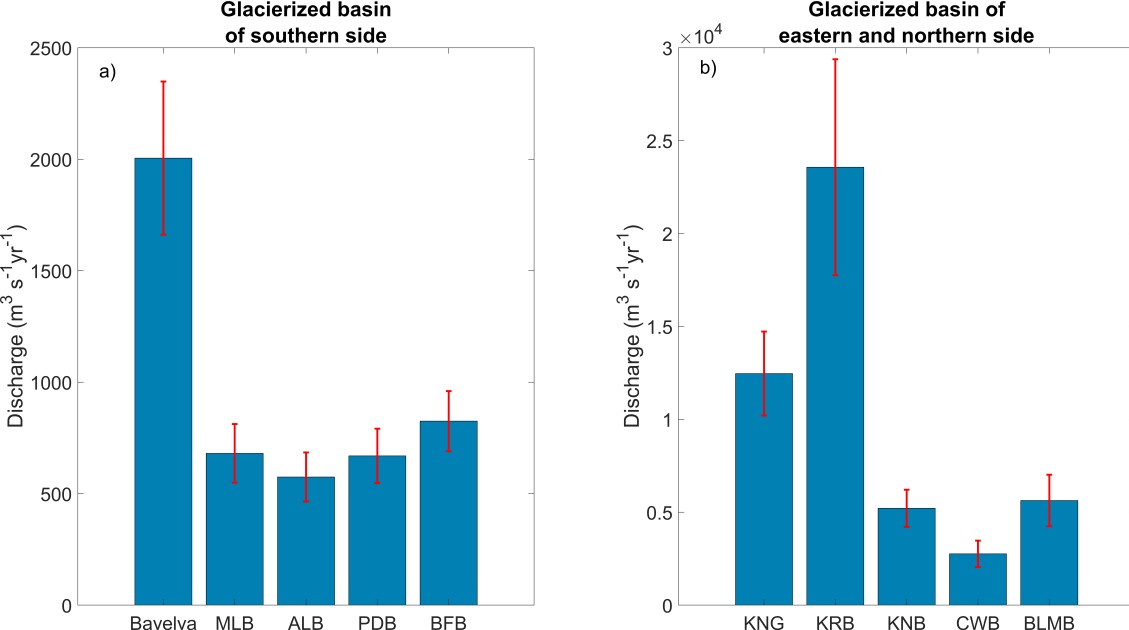

**Figure 6.** Total annual freshwater volume flux averaged over 2013-16 from a) the five largest (area >10 km$^2$) land-terminating glaciers on the southern side of the fjord (left to right: outlets from west to east in Fig. 3a) and b) from the five tidewater glaciers on the east and northern part of the fjord left to right: outlets from east to west in Fig. 3a. Discharge from tidewater glaciers is shown for hydraulic head corresponding to $k = 0.8$. Kronebreen contributes around 39% of the total freshwater to the fjord. Red lines indicate the standard deviation of discharge over the period.



**Table A1.** Area($km^2$) of five tidewater glacier drainage basins for different $k$ values: Kongsvegen (KNG), Kronebreen (KRB), Kongsbreen (KNB), Conwaybreen (CWB), and Blomstrandbreen (BLMB). For $k$ values between 0 and 0.4, KRB and KNG drains together, and the corresponding area is shown separately.

| $k$ | KNG | KRB | KNG+KRB | KNB | CWB | BLMB |
|---|---|---|---|---|---|---|
| 0.1 | - | - | 344 | 596 | 59 | 105 |
| 0.2 | - | - | 855 | 86 | 59 | 107 |
| 0.3 | - | - | 851 | 85 | 59 | 107 |
| 0.4 | - | - | 855 | 85 | 60 | 114 |
| 0.5 | 210 | 655 | - | 78 | 60 | 115 |
| 0.6 | 211 | 659 | - | 76 | 61 | 115 |
| 0.7 | 212 | 664 | - | 77 | 61 | 116 |
| 0.8 | 212 | 668 | - | 75 | 62 | 116 |
| 0.9 | 213 | 677 | - | 77 | 63 | 116 |
| 1 | 214 | 670 | - | 76 | 65 | 116 |





**Figure A1.** Simulated average grid-cell surface runoff for the summer months (a) June, b) July, c) August and d) September), for the period 2013-2016.





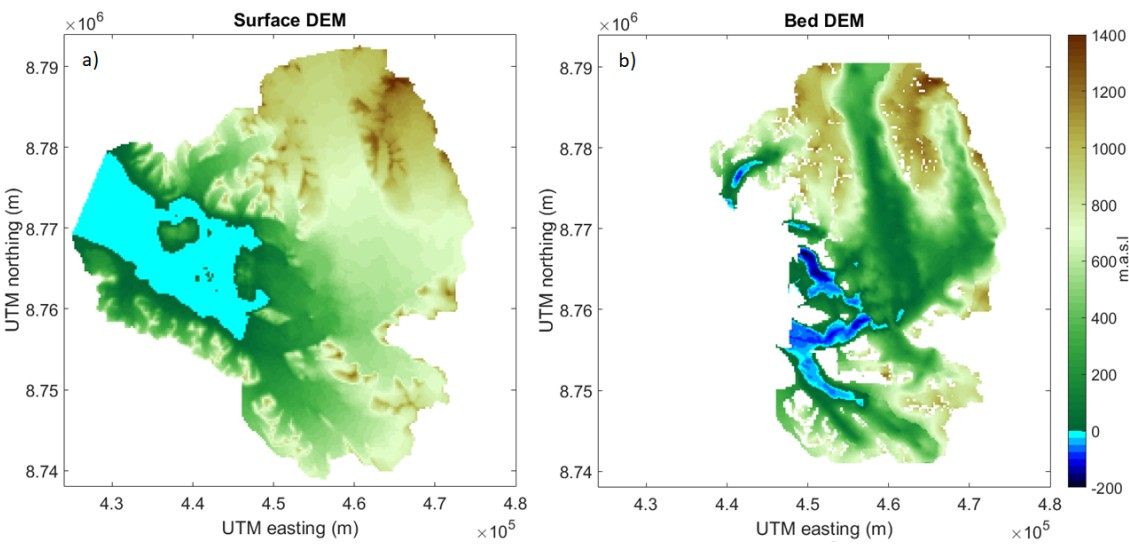

**Figure A2.** a) Surface DEM of Kongsfjord basin, and b) Subglacial DEM of the tidewater glaciers of Kongsfjord basin (Lindbäck et al., 2018). Gaps indicate exposed bedrock and nunataks. Note that in the subglacial DEM only tidewater glaciers are shown as hydraulic potential is calculated only for these glaciers.



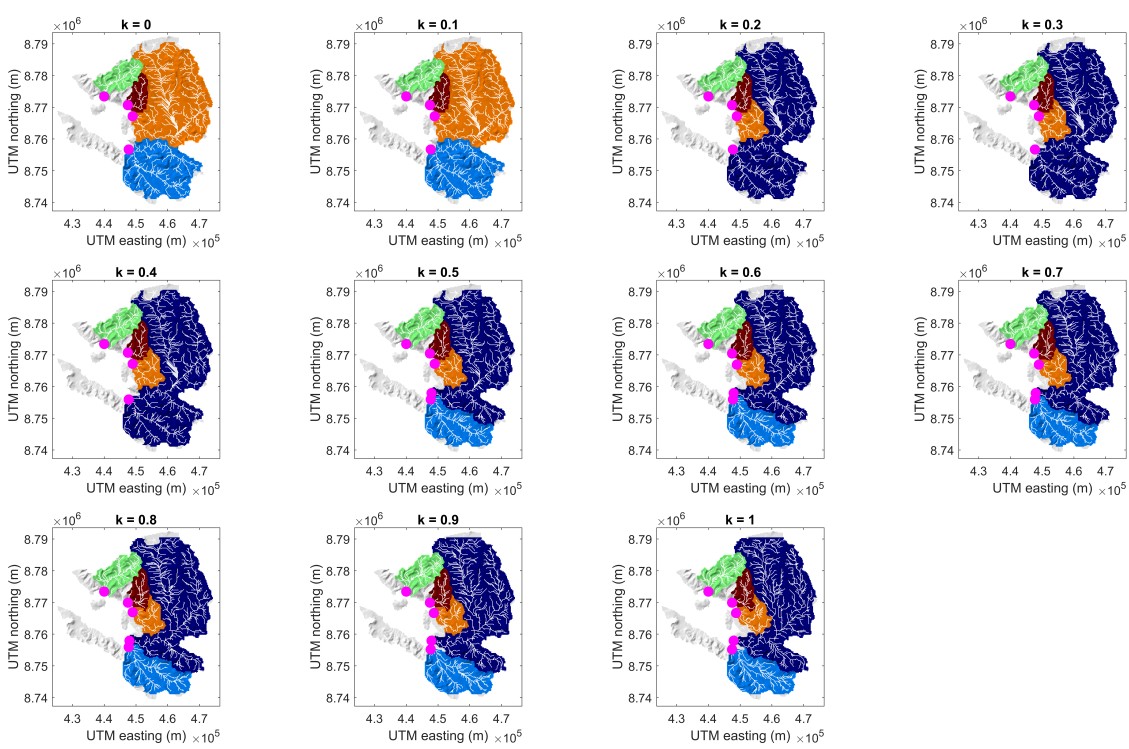

**Figure A3.** Drainage delineation of tidewater glacier drainage basins for different *k* values for 250 m resolution DEM. Drainage basins are named with respective glaciers. Kronebreen basin is marked with blue, Kongsbreen basin with orange, Kongsvegen basin with cyan, Conwaybreen basin with maroon, and Blomstrandbreen basin with green. Same color convention is used in subsequent figures. Basin outlets are shown with magenta circle.


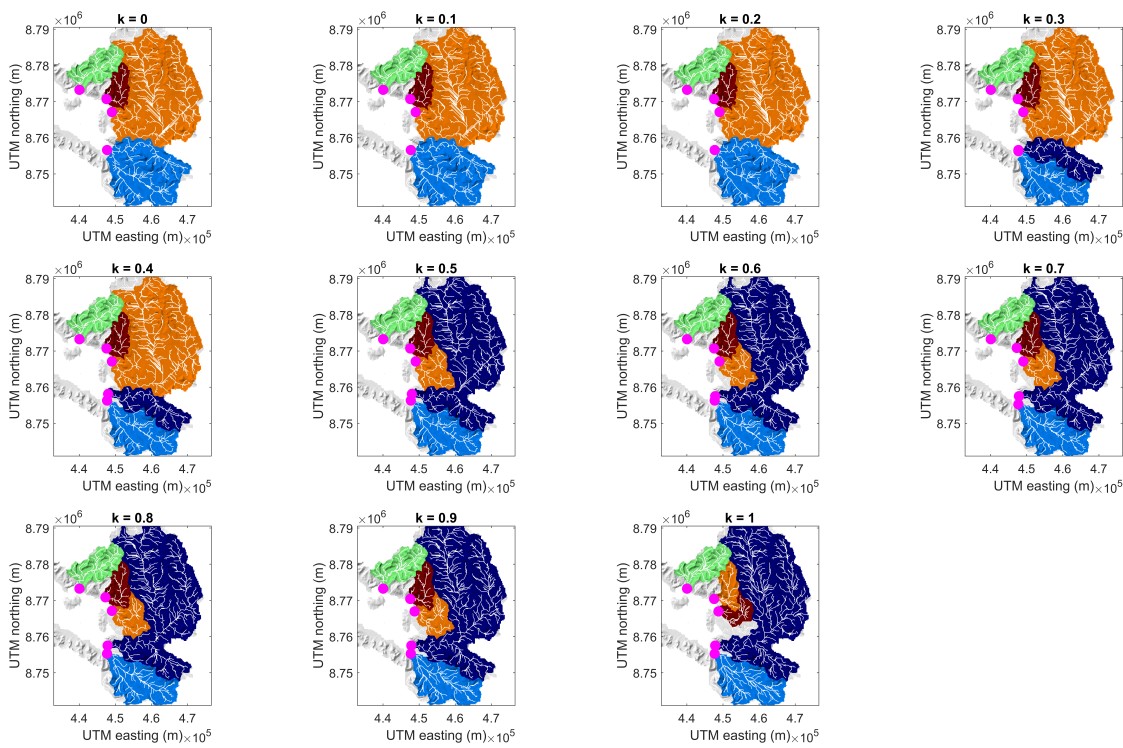

**Figure A4.** Drainage delineation of tidewater glacier drainage basins for different *k* values for 150 m resolution DEM.



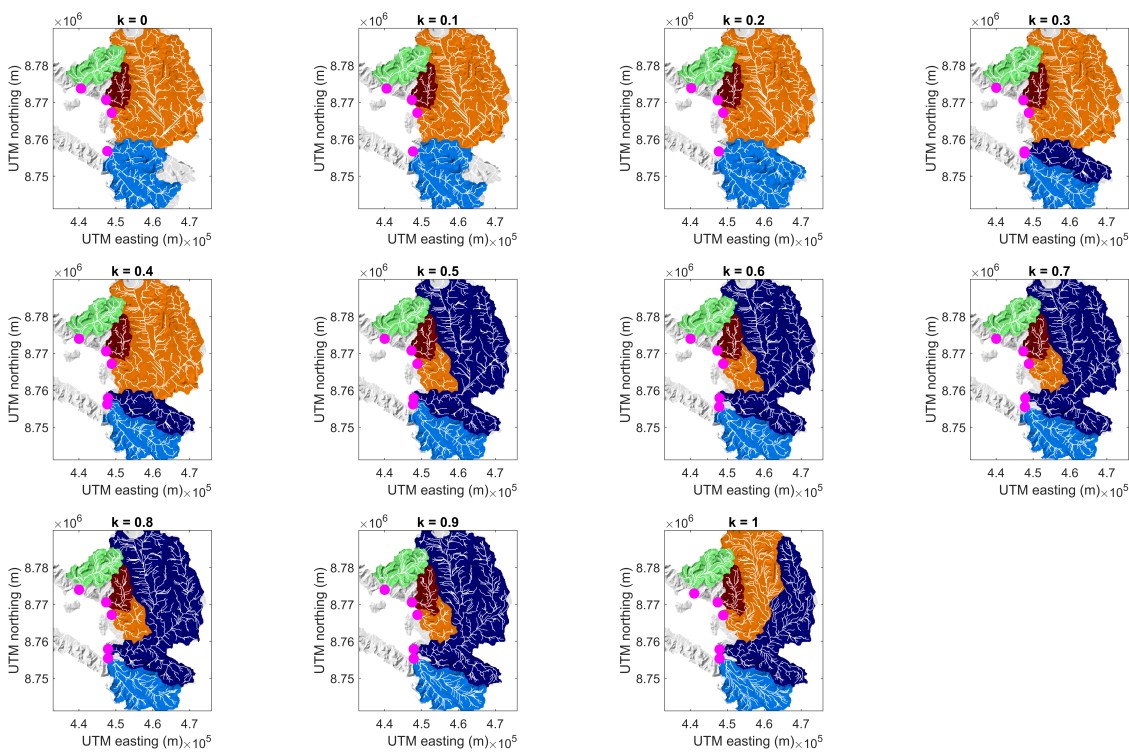

**Figure A5.** Drainage delineation of tidewater glacier drainage basins for different *k* values for 100 m resolution DEM.



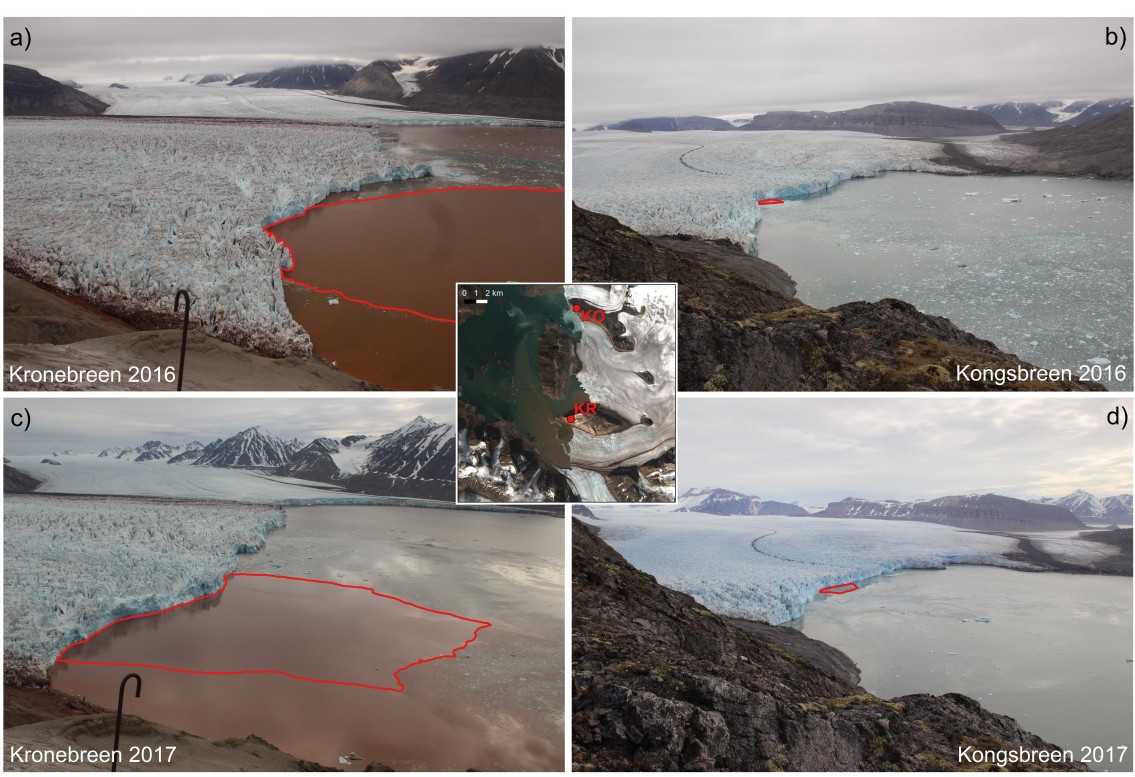

**Figure A6.** Time-lapse camera images of Kronebreen (KRB) and Kongsbreen (KNB) glacier fronts with plume areas marked with red polygons. a) and b) are the images from 5 August 2016, and c) and d) are the images from 17 July 2017. Inset is the Sentinel-2 satellite image over Kongsfjord on 10 July 2016. KR and KO are the camera locations facing KRB and KNB glacier fronts, respectively.





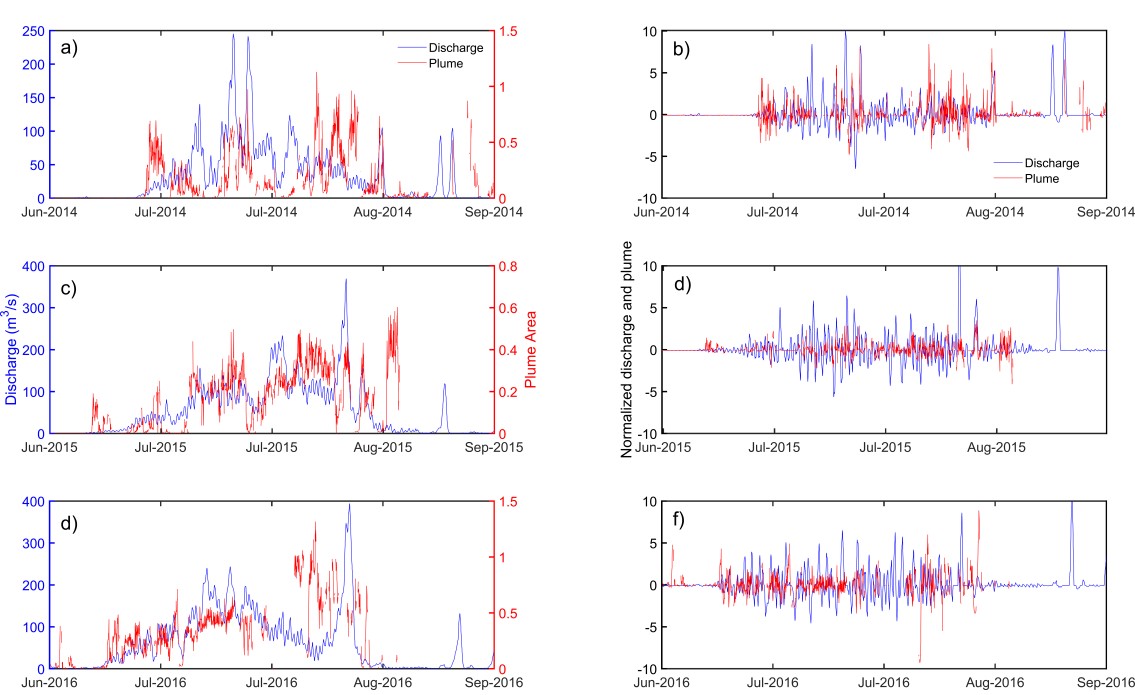

**Figure B1.** Observed plume area extent and no-delay discharge for 2014, 2015 and 2016 (left panel) and corresponding high-frequency component (right panel).





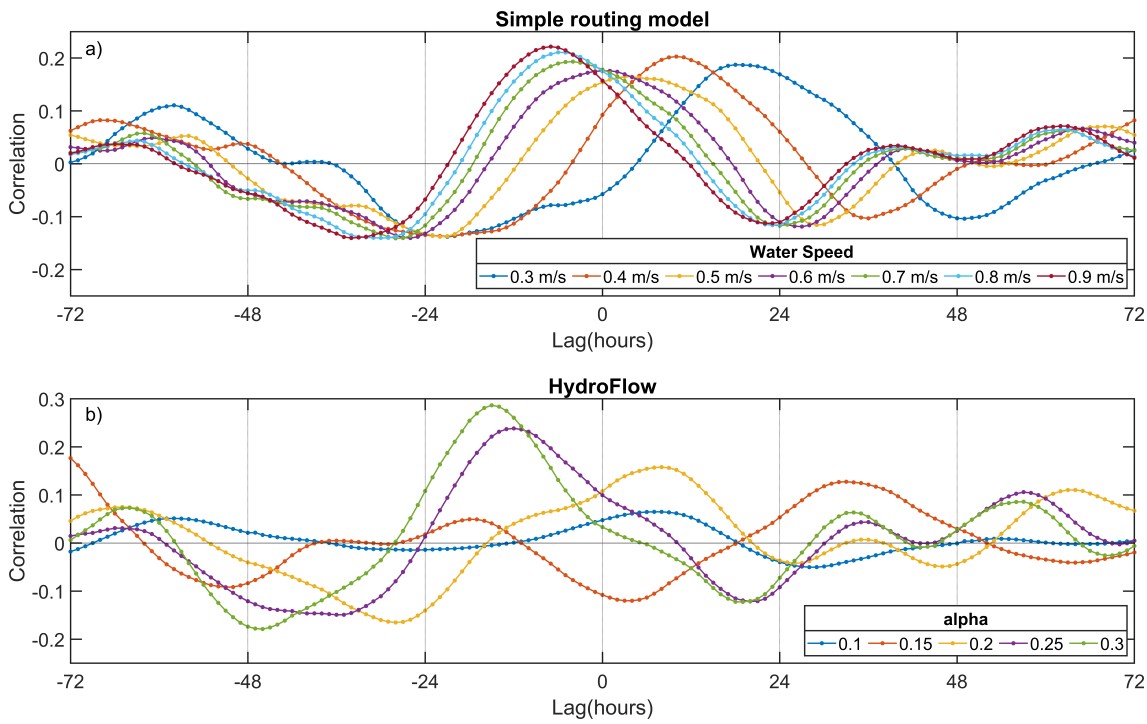

**Figure B2.** Cross-correlation between the high frequency of Kronebreen plume area extent and discharge hydrograph for different values of a) wave speed in the simple routing model; and b) $\alpha$ in HydroFlow.





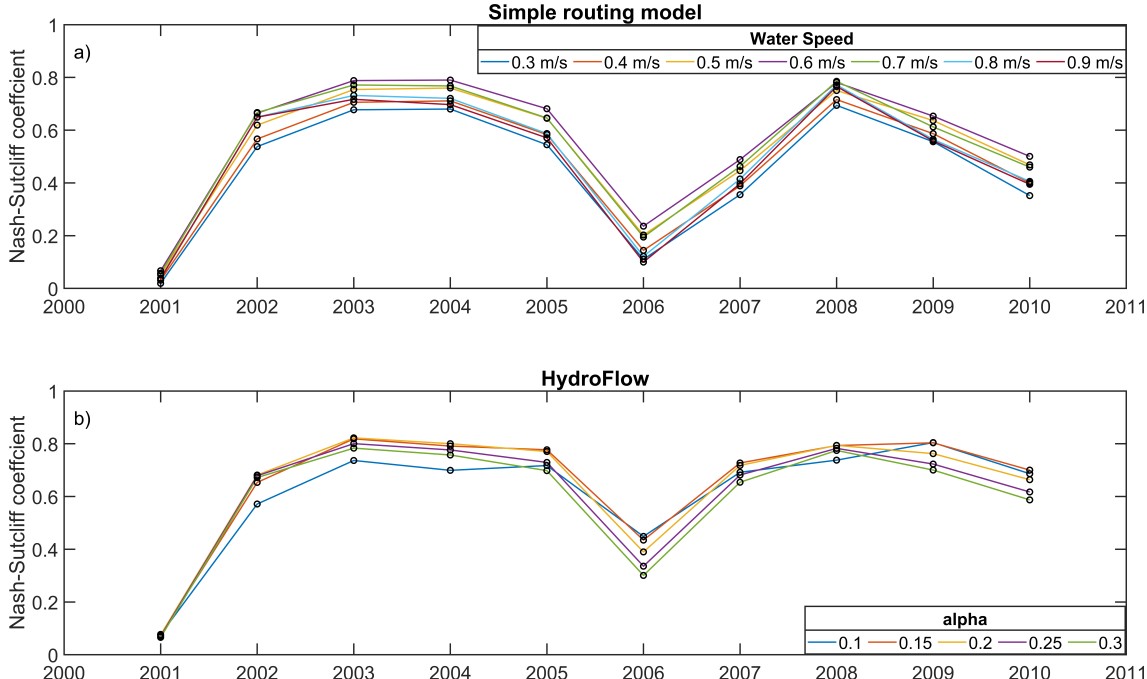

**Figure B3.** Nash Sutcliffe coefficient for different values of the a) wave speed of simple routing model; b) $\alpha$ of HydroFlow for Bayelva. Note that the Bayelva measurements are carried out automatically and there are some errors due to ice build-up and sedimentation at the sensor in certain years. Also, the automatic system sometimes fails to measure initial low discharge, as often sensors may remain partially or completely ice-covered by that time.