# Peer review of "Hydrology and runoff routing of glacierized drainage basins in the Kongsfjord area, northwest Svalbard"

_The Cryosphere, 2020_

## Referee Comment (RC1) · Anonymous Referee #1 · 28 Oct 2020

1. General comments:

This paper presents numerical experiments on meltwater discharge from tidewater and land-terminating glaciers in the Kongsfjord basin in Svalbard. Meltwater runoff was computed by an energy balance and snow process coupled model, which was previously published by the authors (Pramanik et al., 2018). The results of the previous paper are used in this study to investigate water flow through the glaciers and discharge into the fjord. Two different runoff routing models were applied for the glaciers in the basin to obtain time series of discharge from glacier front. Experimental results are presented in terms of flow paths and drainage basin, as well as time series of glacial

discharge (hydrograph) from 2013 to 2016. I enjoyed the readable text and carefully prepared plots. Discharge from tidewater glaciers is drawing attention because of its importance in glacier/ice sheet mass loss and for the interaction of glaciers and the ocean. The authors tackle this problem by applying runoff routing models for a relatively well studied glacier basins in Svalbard, where a long-term proglacial discharge and plume observations are available. Results are interesting and potentially important to understand hydrology of glaciers under a similar setting. A weakness of the study is poorly constrained model parameters. This is critical because validation of the model output is only possible for the land-terminating glacier, where proglacial discharge data are available. Modeled discharge from a tidewater glacier is compared with plume area, but it is insufficient to optimize the model. Because of this shortcoming, it is difficult to assess how realistic the presented results are. Accordingly, discussion of the experimental results is pretty weak and the authors failed to draw important conclusions. In my opinion, more rigorous conclusions are required for a paper published in The Cryosphere. I see the value of the experiment and potential importance of the result, thus encourage the author to perform more careful experiment and writing. In my opinion, the paper will be substantially improved by setting clear objectives of the study, designing experiments to overcome the parameter uncertainties, and analyze experimental results to demonstrate the importance and implication of the study. I list my major concerns below, which are followed by specific comments.

2. Major concerns:

(1) Drainage basin analysis It is interesting to see the drastic change in the drainage basin boundaries, depending on the choice of the parameter "k". I wonder if you can enhance this finding by more detailed presentation and analysis. For example, area of each drainage basin can be plotted against "k", so that quantitative analysis is possible for the impact of "k" on the drainage from each glacier. I am also interested in the mechanism of such migration of the drainage boundary. If you focus the region of "subglacial water piracy" and explain the process in terms of bed/ice geometry, you may

be able to generalize such finding for future research. Please also discuss this finding with an attention on surface water production and water transfer from the surface to the bed. Even if a large area in the upper reach switches to another drainage basin, the influence on the glacier discharge is small in case the area is above the percolation zone. because melt is small and do not penetrate to the bed.

(2) Discharge hydrograph I understand that obtaining a hydrograph is an important goal of this study. The reconstruction of hydrograph is successful for the land terminating glacier (Fig. 5). In contrast, results for tidewater glaciers are not reliable. It is not clear how parameters were tuned for the tidewater glaciers and the validation of the results is not convincing (Fig. 4). Further, parameter settings are very simple, as represented by "k" assumed as uniform in time and space. Therefore, hydrographs for tidewater glaciers are questionable, and uncertainty is unclear. My suggestion is to perform sensitivity tests and evaluate the uncertainties in the results. By taking various values of k, $\alpha$, and water speed, uncertainty can be evaluated for the discharge and presented as a band in Figs 4 and 5. Please also discuss Fig. 4 in terms of agreement between the discharge and plume area. Frankly speaking, I do not see "agreement" in the plot.

(3) Model parameters Parameters uniformly distributed in space and time are very crude assumptions. Significant spatial variations are expected for "k", and it changes over a year particularly in the ablation area. Water flow through a glacier consists of complex processes, thus speed of water movement varies in time and space. Moreover, processes involved in water movement after runoff is given by the melt-snow model is not very clear. Do you assume water drains straight down to the bed? I believe the time required for such process is highly uncertain and variable. Taking all these uncertainties into consideration is not possible, thus some degree of simplification and assumptions are necessary for this study. Nevertheless, I think the treatment of the parameters is too simplistic. In fact, a large portion of Discussion is allocated to describe such short comings. I encourage the authors to perform sensitivity test and provide more rigorous discussion on the model uncertainty.

(4) Objective of the study My question that forms the background of the comments listed above is "what is the objective of the study?". The abstract suggests "delay in discharge" is the main point of the study (line 2). In the end of Introduction, "drainage delineations" and "subglacial network" are raised as the purpose of the modeling (Line 49). Judging from the presented result, delineation of the drainage basin is worth to highlight. However, I am not sure if the study achieved accurate quantification of the delay (Fig. 3b) and what is new about the subglacial network. My suggestion is to define clear study goals. Experimental design, data analysis and presentation should be optimized to achieve them. It is not bad idea to place the focus on the subglacial drainage basin (Fig. 2) and hydrograph (Figs 4 and 5). Setting clear goals of the experiment should guide you to design numerical experiments and data analysis necessary to draw conclusions.

3. Specific comments:

Line 23: "minimal effect on the local circulation" » Freshwater discharge to the fjord surface has an effect to enhance stratification, isn't it?

Line 47: "The first order, Water. . ." » water

Line 47: "determined by bed topography and basal water pressure" » What about ice thickness?

Line 59–64: Please refer to Fig. 1 to explain the study site.

Line 62: "Warm water inflow . . ." » This sentence is not clear. Please be more specific in time rather than "in recent years" and "until recently". Also not clear what happens "since 2006".

"Line 69": "terminus depth" » Not clear if this refers to the ice thickness or fjord depth.

Line 71–75: Because this region is relatively well studied, can you describe more about previous works in this region?

Line 85: "Runoff from a Coupled . . ." » coupled

Line 94: I wonder why you write "interested in discharge over the period 2010–2016" because Figs 4–6 cover only 2013–2016.

Line 120–121: I understand that you assume supraglacial discharge (no basal hydrology) at land-terminating glacier. Please clearly state so if this is correct.

Line 128: Something is wrong with the unit (hyphen and italic). Please check throughout the paper.

Line 134: ". . . water moves according to the bed topography alone." » And ice thickness?

Line 152: What kind of "distance"? Distance from the point on the meltwater production to the glacier front along the drainage path? Can you reword "water wave speed" by "water movement speed" (or water flow speed) as in Line 155? "Wave speed" can be something different, I think.

Line 166: Please consider writing "peak" instead of "maximum".

Line 189: Liston and Mernild (2012) applied the model to a land-terminating glacier. Can you validate the use of the same parameters for tidewater glaciers? For example, the parameter k(t) in Equation (4) in Liston and Mernild (2012) is critically important and dependent on glacier conditions.

Line 206: "Hotedahlfonna and Isachsenfonna" » I understand these are names given on the upper regions of Kronebreen and Kongsbreen. It is confusing to use four names for two glaciers. Cay you reword them with something like "accumulation areas of Kronebreen and Kongsbreen"? It helps readers.

Line 210: "(Fig. A4, A5)" » Can you refer to Fig. 2a instead of the supplementary figures?

Line 215: "For k values between 0.5 and 1, . . ." » Can you refer to Fig. 2d for this

sentence?

Line 226–232: I find the described procedure of the parameter tuning is highly uncertain. The choice of alpha=0.15 sounds subjective. Please consider sensitivity tests instead of fixing the parameters.

Line 235–236: I do not understand the basis to use k=0.8. Why "therefore"?

Line 242: What is the definition of "discharge days"?

Line 243: "late discharge events in 2016" » Are you referring to the peaks in September? Please specify.

Line 251: "We also compare peak discharge between . . ." » Do you compare "timing" of the peaks?

Line 265–270: Please refer to Fig. 2 for the text, i.e. Fig. 2d for the first sentence and Figs 2b and c for the 3rd sentence.

Line 265: Once again, it is confusing to use "Holtedahlfonna and Isachsnfonna".

Line 308–310: What kind of process do you have in your mind about "runoff from basal melting"? Do you assume basal melt events in early summer and in late autumn? Any literature about it?

Line 324–326: I encourage the authors to discuss more about the implication of your study for circulation and biology in the fjord.

Line 380: Space is missing in the unit "ms$-1$".

Figure 6: As far as I understand, this kind of data can be obtained basically with the melt-snow model, but the drainage from tidewater glacier basins is influenced by the water piracy. Can you discuss more about this point to highlight the importance of your study? On Fig. 6b, I would compare this result with that expected by supraglacial drainage as well as those obtained by other values of the parameter "k".

[Figure]

[Figure]

---

## Referee Comment (RC2) · Anonymous Referee #2 · 29 Oct 2020

This study examines the hydrology and subsequent freshwater discharge of the glaciers of Kongsfjord, Svalbard. Results of a previous modelling study (same first author) of surface melt and runoff are used as input for a simple and a more complicated routing model, and discharge hydrographs for each outlet are produced.

Overall I found the figures of this paper to be clear and well presented, and the explanations of the previously published methods utilized here to be well explained, but the explanations of the new science done here and the conclusions drawn require significant work before I could recommend this paper for publication. Although the hydrograph outputs are of interest to the field, it was not clear to me exactly what question

the authors were trying to answer with this study.

Section 5.2 is the key area that is currently lacking in clarity. The choices of these parameter values determine the main output of the study, but I did not find clear justification for the choices made. Furthermore, the choice of why the HydroFlow model is used here seems unclear.

Line by line comments:

Line 10: Is the fact that larger glaciers produce more runoff than smaller ones really a key finding to include in the abstract? Was this not known before the study? Or was it influenced by the water piracy?

Line 18: Are there no supraglacial channels here?

Line 42: Leeson et al 2012 for supraglacial routing (if appropriate here)

Line 47: Capitalized "Water"

Line 60: Is the Pramanik 2019 study relevant here?

Line 63-4: The wording here is confusing. Is there sea ice now? Can you be more specific with exact dates the sea ice was not present.

Line 94: You mention 2010-2016 but the work presented here mainly starts at 2013?

Line 98-100: (and similarly in the Appendix) Why was the resampled grid so low resolution? When you had such a high resolution DEM did you not test your simulations with much higher surface resolutions? What was the justification for making the resampled grid so coarse?

Line 139: Can the grid cells that don't receive any flow still produce in-situ melt (that may then travel outside of the boundary)?

Line 144: If the hydraulic potential is so sensitive to this why not try lower DEM resolutions?

[Figure]

Line 146: Can you quantify the results of the Monte Carlo simulations here? The description in the appendix seems subjective.

Line 152: What do you mean by "distance" here? The distance from production to discharge? Or between grid cells? I'm also a bit confused what you mean by water wave speed, is this the same as flow speed? At some points you just refer to "water speed" later on in the text, is this the same thing?

Line 159: What is the justification for a uniform speed? Will this not be significantly affected by channel size, presence of firn, etc?

Line 162: How did you determine that this speed was optimal? This seems quite key to not justify in more detail.

Line 173: Can you define the Nash-Sutcliffe coefficient and give a brief explanation of how it is used. This will likely be unfamiliar to many readers.

Line 191: What is the justification for the choice of using HydroFlow here? If subglacial hydrology is important was this model really the right choice if you're having to use similar calibrations to the simple model? It's unsurprising in this case that the results don't differ so much.

Line 227: "Visual inspection" doesn't suggest to me that this key choice has been made rigorously.

Line 235-263: I don't follow your use of "therefore" here. I can't see justification for the choice of k=0.8.

Line 250: The reader needs knowledge of NSC to understand these numbers- why do they show good agreement?

Line 279: Can you elaborate on how the simulations and data support this, or we just have to take your word for it.

Line 282: How do the DEM resolutions compare between the studies?

Line 317: Have you tried your simulations at a sub-hourly timescale? (If not, why not?)

Line 324-326: Can you elaborate on exactly how your conclusions will affect this?

Line 380: How was the choice of cut-off made?

Line 383: Can you use a number instead of "good" which is subjective.

Figure 4: The grey here is hard to see.

---

## Author Comment (AC1) · 11 Jan 2021

**Response:** We thank the reviewer for taking the time to provide such a thoughtful and thorough review. We greatly appreciate the reviewer's suggestion to provide discussions on the region of subglacial water piracy and setting clear objectives of the study. We have addressed all of your points, and list them alongside your review.

*R1: 1. General comments:*

*This paper presents numerical experiments on meltwater discharge from tidewater and land-terminating glaciers in the Kongsfjord basin in Svalbard. Meltwater runoff was computed by an energy balance and snow process coupled model, which was previously published by the authors (Pramanik et al., 2018). The results of the previous paper are used in this study to investigate water flow through the glaciers and discharge into the fjord. Two different runoff routing models were applied for the glaciers in the basin to obtain time series of discharge from glacier front. Experimental results are presented in terms of flow paths and drainage basin, as well as time series of glacial discharge (hydrograph) from 2013 to 2016. I enjoyed the readable text and carefully prepared plots. Discharge from tidewater glaciers is drawing attention because of its importance in glacier/ice sheet mass loss and for the interaction of glaciers and the ocean. The authors tackle this problem by applying runoff routing models for a relatively well studied glacier basins in Svalbard, where a long-term proglacial discharge and plume observations are available. Results are interesting and potentially important to understand hydrology of glaciers under a similar setting. A weakness of the study is poorly constrained model parameters. This is critical because validation of the model output is only possible for the land-terminating glacier, where proglacial discharge data are available. Modeled discharge from a tidewater glacier is compared with plume area, but it is insufficient to optimize the model. Because of this shortcoming, it is difficult to assess how realistic the presented results are. Accordingly, discussion of the experimental results is pretty weak and the authors failed to draw important conclusions. In my opinion, more rigorous conclusions are required for a paper published in The Cryosphere. I see the value of the experiment and potential importance of the result, thus encourage the author to perform more careful experiment and writing. In my opinion, the paper will be substantially improved by setting clear objectives of the study, designing experiments to overcome the parameter uncertainties, and analyze experimental results to demonstrate the importance and implication of the study. I list my major concerns below, which are followed by specific comments.*

**Response:** We agree that the parameter choice of simple routing model is very simplistic in nature. To have a robust estimate of discharge, a detail understanding of subglacial hydrology is required with complex modelling and observational data. At present, there is not enough understanding about the subglacial hydrology of the glaciers of this region. Our aim is to simulate discharge at all the outlets of Kongsfjord basin, and with the absence of enough observational subglacial data, we applied a simple approach to give a coarse estimate of discharge. However, if these discharge data are used for daily time period, then we argue that the discharge will have less uncertainty considering the area of the basin and associated delay from the farthest grid cell of the basin.

The runoff produced at the surface takes some time to reach the glacier front which is termed here as delay, and water storage, if any, is considered as negligible in comparison to discharge. Therefore, the sum of discharge and sum of runoff should be equal over a season, however, the discharge hydrograph with and without delay should be different.

We agree that the delay depends on many factors, such as channel shape, bed property, etc., and these are unknown to us. Therefore, we used a single parameter box model to simulate discharge hydrograph. A further approach can try to parameterize the processes governing the delay with a higher-order model, however, that is beyond the scope of this study. Moreover, a higher-order model, such as GlaDS, has more number of parameters, which, in the absence of observation, would add further uncertainty.

Furthermore, we would like to mention that in a recent paper by Mankoff et al., 2020, a similar approach was taken to calculate subglacial drainage delineations of Greenland basins. In that paper, the discharge hydrographs are being simulated for all the outlet glaciers in Greenland, where the transition from runoff to discharge was considered instantaneous. In this paper we took a further measure to incorporate delay to runoff in a coarse way. We argue that with the paucity of observational data, simple and conceptual routing model can serve the purpose of discharge calculation, and the robustness increases with smaller basin and with lower-temporal resolution.

**R1:** 2. Major concerns:

**R1:** (1*) Drainage basin analysis:  It is interesting to see the drastic change in the drainage basin boundaries, depending on the choice of the parameter "k". I wonder if you can enhance this finding by more detailed presentation and analysis. For example, area of each drainage basin can be plotted against "k", so that quantitative analysis is possible for the impact of "k" on the drainage from each glacier. I am also interested in the mechanism of such migration of the drainage boundary. If you focus the region of "subglacial water piracy" and explain the process in terms of bed/ice geometry, you may be able to generalize such finding for future research. Please also discuss this finding with an attention on surface water production and water transfer from the surface to the bed. Even if a large area in the upper reach switches to another drainage basin, the influence on the glacier discharge is small in case the area is above the percolation zone. because melt is small and do not penetrate to the bed.*

**Response:** Thank you for your suggestion. We agree that the area of the drainage can be plotted with k, but please note that the area of the drainage basins changes drastically for certain k values (e.g., k = 0.1 and k = 0.4). For rest of the k values the drainage area changes are very less ($\sim$ 2km$^2$) to be captured in a plot. Therefore, instead of showing it in a figure, we presented the areas in table (Table A1.)

It is a good point to include a discussion on the region of subglacial water piracy (we would term it as the switching zone). We would conduct a comprehensive analysis of this switching zone. In the revised manuscript, we will make a separate section of switching zone in the discussion where we would discuss about the areas of switching zone and provide how possible changes occurring in this area would lead to changes in subglacial drainage delineations.

It is a fact that if the area is above percolation zone, the water piracy would have smaller effect. In this study, the area where the switching occurs is situated in the upper ablation zone of the glaciers (Line 217-221). Therefore, we mentioned that the water piracy would affect discharge in these two outlets (KRB and KNG) only in peak summer and not in early summer/late autumn.

**R1:** (2*) Discharge hydrograph: I understand that obtaining a hydrograph is an important goal of this study. The reconstruction of hydrograph is successful for the land terminating glacier (Fig. 5). In contrast, results for tidewater glaciers are not reliable. It is not clear how parameters were tuned for the tidewater glaciers and the validation of the results is not convincing (Fig. 4). Further, parameter settings are very simple, as represented by "k" assumed as uniform in time and space. Therefore, hydrographs for tidewater glaciers are questionable, and uncertainty is unclear. My suggestion is to perform sensitivity tests and evaluate the uncertainties in the results. By taking various values of k, alpha, and water speed, uncertainty can be evaluated for the discharge and presented as a band in Figs 4 and 5. Please also discuss Fig. 4 in terms of agreement between the discharge and plume area. Frankly speaking, I do not see "agreement" in the plot.*

**Response:** In the absence of any discharge measurement of tidewater glaciers, we assumed plume as proxy to discharge. We agree that this assumption is also crude as many factors affects plume area, however, one major controlling factor of plume emergence is subglacial discharge. Here, we tried to match the high-frequency of plume area signal with the high-frequency of discharge hydrograph for different alpha/water speed, and we optimized the values from there (Normalized cross correlation).

We agree that it is difficult to optimize the wave speed and many of the wave speed gives similar results. Therefore, instead of finding one single wave speed value, we will use the range of prescribed uniform wave speed and presented the discharge hydrograph as a band, instead of a single line.

For k sensitivities, we did Monte-Carlo simulations with randomly varying k value spatially. For different k values, we only can find certain drainage basin changes between Holtedahlfonna and Isachsenfonna. Therefore, uncertainties in k values would raise only two possibilities, which we discussed (L205-210).

**R1:** (3) *Model parameters:  Parameters uniformly distributed in space and time are very crude assumptions. Significant spatial variations are expected for "k", and it changes over a year particularly in the ablation area. Water flow through a glacier consists of complex processes, thus speed of water movement varies in time and space. Moreover, processes involved in water movement after runoff is given by the melt-snow model is not very clear. Do you assume water drains straight down to the bed? I believe the time required for such process is highly uncertain and variable. Taking all these uncertainties into consideration is not possible, thus some degree of simplification and assumptions are necessary for this study. Nevertheless, I think the treatment of the parameters is too simplistic. In fact, a large portion of Discussion is allocated to describe such short comings. I encourage the authors to perform sensitivity test and provide more rigorous discussion on the model uncertainty.*

**Response:** We agree that the assumption of uniform distribution of model parameters is crude assumptions, and there is significant variations. In the manuscript, we mentioned that k varies spatially and temporally. To better address this, we conducted rigorous sensitivity analysis (Appendix L356-374). We varied k randomly for 10000 Monte-Carlo runs and calculated subglacial drainage delineations and discussed our findings (Appendix L365-369).

The energy-balance model calculates runoff, that is melt water reaching snow-ice interface after percolation. Yes, here we assume that the runoff instantly reaches bed without any time-delay.

We also believe that there are some number of uncertainties, however, it is not possible to take all those uncertainties into consideration. For subglacial hydrology, we have conducted detailed sensitivity tests to provide a robust estimate. We agree that the treatment of wave speed /alpha parameter in the simple routing model is simplistic in nature. We would like to point out that there is no observational data available from subglacial environment and the use of multiple parameters would further increase the uncertainty of the results. Basically, here we considered that the input (runoff) and output (discharge) is constant and the in between process is a black box which is parameterized with a single parameter. Our simple routing model is apparently a single parameter box model of subglacial water transport without any storage.

We agree that with this approach it is not appropriate to optimize the parameter for tidewater glaciers. Instead, we will calculate the discharge hydrograph for a range of water speed values taken from (Cowton et al., 2013, Slater et al., 2017) provide the discharge hydrograph for tidewater glaciers as a band. A preliminary figure of discharge hydrograph for Kronebreen is provided here (Fig. Res1).

[Figure]

Fig. Res1. Discharge hydrograph for Kronebreen.

Furthermore, we will conduct sensitivity analysis in the switching zone and incorporate that with the routing model to provide discharge hydrograph as a band.

**R1:** (4) *Objective of the study: My question that forms the background of the comments listed above is "what is the objective of the study?". The abstract suggests "delay in discharge" is the main point of the study (line 2). In the end of Introduction, "drainage delineations" and "subglacial network" are raised as the purpose of the modeling (Line 49). Judging from the presented result, delineation of the drainage basin is worth to highlight. However, I am not sure if the study achieved accurate quantification of*

*the delay (Fig. 3b) and what is new about the subglacial network. My suggestion is to define clear study goals. Experimental design, data analysis and presentation should be optimized to achieve them. It is not bad idea to place the focus on the subglacial drainage basin (Fig. 2) and hydrograph (Figs 4 and 5). Setting clear goals of the experiment should guide you.*

**Response:** The main objective of the study is to calculate discharge hydrograph at all the outlets of Kongsfjord basin. To understand water routing of tidewater glaciers, we conducted subglacial hydrology analysis. We will rewrite the introduction with a focus on subglacial drainage basin and hydrograph.

---

## Author Comment (AC2) · 11 Jan 2021

**Response:** We thank the reviewer for the comments and suggestions to further improve the manuscript. We have addressed all your points alongside your review.

*This study examines the hydrology and subsequent freshwater discharge of the glaciers of Kongsfjord, Svalbard. Results of a previous modelling study (same first author) of surface melt and runoff are used as input for a simple and a more complicated routing model, and discharge hydrographs for each outlet are produced.*
*Overall I found the figures of this paper to be clear and well presented, and the explanations of the previously published methods utilized here to be well explained, but the explanations of the new science done here and the conclusions drawn require significant work before I could recommend this paper for publication. Although the hydrograph outputs are of interest to the field, it was not clear to me exactly what question the authors were trying to answer with this study.*

*Section 5.2 is the key area that is currently lacking in clarity. The choices of these parameter values determine the main output of the study, but I did not find clear justification for the choices made. Furthermore, the choice of why the HydroFlow model is used here seems unclear.*

**Response:** We agree that the parameter choice of simple routing model is very simplistic in nature. To have a robust estimate of discharge, a detail understanding of subglacial hydrology is required with complex modelling and observational data. At present, there is not enough understanding about the subglacial hydrology of the glaciers of this region. Our aim was to simulate discharge at all the outlets of Kongsfjord basin, and with the absence of enough observational subglacial data, we applied a simple approach to give a coarse estimate of discharge. However, if these discharge data are used for daily time period, then we argue that the discharge will have less uncertainty considering that the area of the basin and associated delay from the farthest grid cell of the basin.

The runoff produced at the surface takes some time to reach the glacier front which is termed here as delay, and water storage, if any, is considered as negligible in comparison to discharge. Therefore, the sum of discharge and sum of runoff should be equal over a season, however, the discharge hydrograph with and without delay should be different. We agree that the delay depends on many factors, such as channel shape, bed property, etc., and these are unknown to us. Therefore, we used a single parameter box model to simulate discharge hydrograph. A further approach can try to parameterize the processes governing the delay with a higher-order model, however, that is beyond the scope of this study. Additionally, a higher-order model, such as GlaDS, has many numbers of parameters, which, in the absence of observation, would add further uncertainty.

Furthermore, we would like to mention that in a recent paper by Mankoff et al., 2020, a similar approach was taken to calculate subglacial drainage delineations of Greenland basins. In that study, discharge hydrographs are being simulated for all the outlet glaciers in Greenland, where the transition from runoff to discharge was considered instantaneous. In this paper we took a further measure to incorporate delay to runoff in a coarse way. We argue that with the paucity of observational data simple and conceptual routing model can serve the purpose of discharge calculation, and the robustness increases with smaller basin and with lower-temporal resolution.

*Line by line comments:*

*Line 10: Is the fact that larger glaciers produce more runoff than smaller ones really a key finding to include in the abstract? Was this not known before the study? Or was it influenced by the water piracy?*

**Response:** The water discharge is influenced by water piracy for the two big tidewater glaciers. However, it is not the finding of the study that larger glaciers produce more runoff, hence we would remove it from the abstract.

*Line 18: Are there no supraglacial channels here?*

**Response:** We have rewritten the statement as "Meltwater generated at the glacier surface reaches the glacier front after travelling through supraglacial channels, crevasses, moulins, and englacial and subglacial channels, emerging at or near the base of the tidewater glacier front to create buoyant plumes."

*Line 42: Leeson et al 2012 for supraglacial routing (if appropriate here)*

**Response:** We could not find this reference. It would be good for us if you kindly mention the link of the paper. If it is Clason et al., 2012 paper, we agree that it would be appropriate to include here.

*Line 47: Capitalized "Water"*

**Response:** We have corrected it.

*Line 60: Is the Pramanik 2019 study relevant here?*

**Response:** Here we tried to mention the interdisciplinary studies combined with glaciology. Therefore, we feel the reference is relevant here.

*Line 63-4: The wording here is confusing. Is there sea ice now? Can you be more specific with exact dates the sea ice was not present.*

**Response:** We will add that in the revised manuscript.

*Line 94: You mention 2010-2016 but the work presented here mainly starts at 2013?*

**Response:**  Corrected it.

*Line 98-100: (and similarly in the Appendix) Why was the resampled grid so low resolution? When you had such a high-resolution DEM did you not test your simulations with much higher surface resolutions? What was the justification for making the resampled grid so coarse?*

**Response:** Yes, there is high-resolution surface DEM available, but the highest resolution of bed DEM is 150 m (Lindback et al., 2018). The spatial resolution of runoff is 250 m (Pramanik et al., 2018).  Therefore, we used 150m, 250m and 100 m resolution DEMs for drainage delineations and 250 m resolution DEM for routing model.
Furthermore, sensitivity analysis with different DEMs do not show much change in subglacial drainage delineations.

*Line 139: Can the grid cells that don't receive any flow still produce in-situ melt (that may then travel outside of the boundary)?*

**Response:** Yes, the grid cells that do not receive any flow can produce in-situ melt, but they would follow the slope of the basin. First basin boundaries are demarcated from the DEMs. Therefore, the in-situ melt of edge grid cells must follow the slope of the basin.

*Line 144: If the hydraulic potential is so sensitive to this why not try lower DEM resolutions?*

**Response:** We have conducted the analysis with 100m, 150m, and 250m DEM resolutions. The highest resolution of bed DEM is 150 m, whereas the runoff resolution is 250 m. We resampled the runoff, surface DEM and bed DEM to different resolutions and conducted the analysis.

*Line 146: Can you quantify the results of the Monte Carlo simulations here? The description in the appendix seems subjective.*

**Response:** In the method section we mentioned that we have conducted Monte-Carlo simulations for subglacial drainage delineations. The Monte-Carlo simulation results are mentioned in the results section of the main article (L210-214) while a detailed discussion is provided in the appendix.

*Line 152: What do you mean by "distance" here? The distance from production to discharge? Or between grid cells? I'm also a bit confused what you mean by water wave speed, is this the same as flow speed? At some points you just refer to "water speed" later on in the text, is this the same thing?*

**Response:** Here 'distance' means the length of the stream for each grid cell to its corresponding outlet point. So, it is the distance from production to discharge.
Water wave speed is same as flow speed. We will now correct it throughout the text and use it as flow speed.

*Line 159: What is the justification for a uniform speed? Will this not be significantly affected by channel size, presence of firn, etc?*

**Response:** Yes, the flow speed is significantly affected by different things, which are difficult to quantify/measure to finally get a robust estimate as most of the subglacial parameters are unknown to us. Therefore, we took a simplistic approach by considering a uniform wave speed (Cowton et al., 2013, Slater et al., 2017). Thereby, we mention a simple or conceptual single parameter routing model. A further approach can try to parameterize the processes governing the delay with a higher-order model (such as GlaDS), however, that is beyond the scope of this study.
Furthermore, we have mentioned these shortcomings of the simple routing model in the discussion section.

*Line 162: How did you determine that this speed was optimal? This seems quite key to not justify in more detail.*

**Response:** In the absence of actual discharge measurement, we considered discharge and plume area as two different signals and tried to match them by varying the flow speed. In the revised version, we will mention that finding an optimum wave speed is too much detail with this simple model. Therefore, we will present the discharge hydrograph as band with the prescribed flow speed range, a preliminary figure for Kronebreen is attached here (Fig. Res1).

[Figure]

Fig. Res1. Discharge hydrograph for Kronebreen.

*Line 173: Can you define the Nash-Sutcliffe coefficient and give a brief explanation of how it is used. This will likely be unfamiliar to many readers.*

**Response:** We agree that Nash-Sutcliff coefficient term can be unfamiliar. We would provide brief explanation of it in the revised manuscript.

*Line 191: What is the justification for the choice of using HydroFlow here? If subglacial hydrology is important was this model really the right choice if you're having to use similar calibrations to the simple model? It's unsurprising in this case that the results don't differ so much.*

**Response:** Our primary aim was to use a conceptual model as well as a physically based model (HydroFlow) to derive discharge hydrograph. HydroFlow is not a fully physically based model for subglacial hydrology, but it works well for supraglacial hydrology. Therefore, we will use HydroFlow for the land-terminating glaciers only where supraglacial hydrology is dominant.

*Line 227: "Visual inspection" doesn't suggest to me that this key choice has been made rigorously.*

**Response:** We agree with the comments and therefore did not try to find an optimum wave speed. Instead, we presented our discharge hydrograph as a band.

*Line 235-263: I don't follow your use of "therefore" here. I can't see justification for the choice of k=0.8.*

**Response**: The subglacial water piracy is evident for a range of k values from 0.5 to 1, and for any of these k values the results will be same. Therefore, considered a median k value. "The hydraulic potential analysis shows a prevalence of water piracy from Isachsenfonna to Holtedahlfonna occurred for k values between 0.5 and 1, therefore, we use the hydraulic head for k = 0.8 (the average of the k value range) to derive water routing." (L234-236)

*Line 250: The reader needs knowledge of NSC to understand these numbers- why do they show good agreement?*

**Response:** We will provide brief information and importance of NSC (Nash-Sutcliff coefficient) in the revised manuscript.

*Line 279: Can you elaborate on how the simulations and data support this, or we just have to take your word for it.*

**Response:** We have done 10000 runs of Monte-Carlo simulations with randomly varying k values and calculated subglacial drainage delineations where we found water piracy from ISF to HDF in 94% cases. Here, by observational data, we mean the plume data at the two tidewater glaciers front (Fig. A6). The observational data and the simulation both support that there is a possibility of subglacial water piracy between ISF to HDF.

*Line 282: How do the DEM resolutions compare between the studies?*

**Response:** We used updated DEM of Lindback et al. 2018, where How et al, 2017 used an earlier version. We suspect that there lies the little discrepancy. However, the discrepancy between two findings is not substantial.

*Line 317: Have you tried your simulations at a sub-hourly timescale? (If not, why not?)*

**Response:** The simulation is done on 6-hourly timescale as the gridded runoff data from Pramanik et al., 2018 was simulated on 6-h timescale.

*Line 324-326: Can you elaborate on exactly how your conclusions will affect this?*

**Response:** As the tidewater glaciers contribute most of the water through different locations in the fjord, depending upon their appearance fjord circulation and ecosystem would evolve in space and time. For example, higher freshwater discharge from glaciers in the inner fjord would increase the possibility of sea ice formation which may affect the foraging of birds. Therefore, an accurate quantification of discharge through different outlets and their relative sizes are important.

*Line 380: How was the choice of cut-off made?*

**Response:** The choice of cut-off is elaborated in the following sentences. "We choose a cut-off frequency to filter out low frequency components and compare the high frequency of plume with the high frequency of discharge data. We did not find good correlation of plume and discharge data for all cut-off frequency. We used different cut off frequencies and

calculated normalized cross-correlation. A good correlation is observed for the cut-off frequency of 59 hours and above. Therefore, we use that as the final cut-off frequency to extract high-frequency signal to calibrate the water speed of the routing model" (L380-384)

*Line 383: Can you use a number instead of "good" which is subjective.*

**Response:** We would provide the correlation/rmse here instead of 'good' in the revise manuscript.

*Figure 4: The grey here is hard to see.*

**Response:** We would use here a corrected figure with hydrograph shown as a band. A preliminary figure is attached here.